# MIXTURE-OF-EXPERTS IN PROMPT OPTIMIZATION

## ABSTRACT

Large Language Models (LLMs) exhibit strong generalization power in adapting to novel tasks when prompted with language instructions and in-context demos. Since this ability sensitively depends on the quality of prompts, various methods have been explored to automate the instruction design process. While these methods demonstrated promising results, they also restricted the output space of the search problem to a demo-free instruction. Such simplification significantly limits their performance, as a single demo-free instruction might not be able to cover the entire problem space of the targeted task due to its complexity. To alleviate this issue, we adopt the Mixture-of-Expert paradigm to divide the problem space into homogeneous regions, each governed by a specialized expert. To further improve the coverage of each expert, we expand their prompts to contain both an instruction and several demos. A two-phase process is developed to construct the specialized expert for each region: (1) *demo assignment*: Inspired by the theoretical connection between in-context learning and kernel regression, we group demos into clusters based on their semantic similarity and assign a cluster to each expert; (2) *instruction assignment*: A region-based joint search is applied to optimize an instruction complementary to the demo cluster for each expert, yielding a synergistic effect. The resulting method, codenamed Mixture-of-Prompts (MoP), outperforms prior art by up to 43% on benchmark NLP tasks.

## 1 INTRODUCTION

Recent advancements in language models have demonstrated a remarkable ability to solve novel tasks described by user instructions (Ouyang et al., 2022; OpenAI, 2023; Touvron et al., 2023; Peters et al., 2018; Devlin et al., 2018; Brown et al., 2020; Wei et al., 2022b). Despite the success, there still exists a substantial gap between user intention and the model's interpretation. Therefore, carefully designed prompts (a.k.a. Prompt Engineering) become an essential ingredient for fully eliciting LLM's superior generalization ability (Alhoshan et al., 2022; Zhao et al., 2021; Liu et al., 2021; Lu et al., 2021; Su et al., 2022; Wang et al., 2022a; Wei et al., 2022a; Yao et al., 2023; Schick et al., 2023; Kojima et al.). However, it usually requires laborious efforts through inefficient trial and error. To automate this process, several recent attempts have shown tremendous potential in utilizing LLMs themselves to design prompts for language generation (Zhou et al., 2022; Pryzant et al., 2023; Chen et al., 2023). Pioneering efforts along this line formulate prompt optimization as a search problem, which aims to find the optimal demo-free instruction based on a set of (input, output) demonstrations for a certain task. While the prompts produced by these methods can outperform human-designed counterparts, restricting the output space of the search problem to a single demo-free instruction significantly limits their problem-solving potential, as the problem space of NLP tasks can be too complex to be covered by a single demo-free instruction.

This paper aims to expand the problem space coverage for automatic prompting by optimizing a Mixture of Prompts (MoP). Our key insight is to adopt the **Mixture of Experts (MoE)** paradigm (Jacobs et al., 1991; Jordan & Jacobs, 1994) to divide the entire problem space into multiple homogeneous regions, each governed by a specialized expert. At inference time, a single expert will be selected to prompt the LLM to answer a new input query. Under the MoE framework, prompt optimization reduces to an **expert assignment problem** that aims to search for the most suitable prompt for each expert, with the goal of **optimizing the performance of their mixture as a whole**.

Another primary improvement proposed in this paper is to expand the prompt for each expert to contain both the instruction and demos jointly optimized for each expert region in the problem space. Intuitively, concrete demos are good at defining fine-grained details and expertise (local information) matching the queries in a local region, whereas the instruction provides a generic ability and

high-level description to solve a task (global information); Hence, they together empower the experts to excel at their problem region. Motivated by this, we adopt a two-phase search algorithm that jointly optimizes a (demos, instruction) pair per expert: We first group all demos into homogeneous clusters, one for each expert, and then search for the best instruction complementary to each demo cluster in a prompt. For the first phase, i.e., demo assignment, we cluster the demos to multiple regions in a semantic embedding space by clustering algorithms. For the second phase, i.e., instruction assignment, we introduce a region-based joint search to find the best instruction to complement the specialty of each expert. Given a new test query, we assign the expert containing the semantically closest demo to it. This method is inspired by the recently established theoretical connection between In-Context Learning and Kernel Regression (Han et al., 2023), which suggests that demos that are semantically closer to a test input tend to perform better at inferring its answer.

We scrutinize the proposed Mixture-of-Prompts (MoP) through extensive empirical study. Our key findings can be summarized as follows: (1). We show that clustering in the embedding space can effectively push semantically similar demos together, obtaining superior ability in solving test points that fall into this region. (2). More experts are not necessarily better: there exists an optimal number of partitions for the problem space. (3). The optimal instruction for each demo split is often distinct, necessitating the joint search of demo and instructions. We further validate the strength of MoP on the commonly used 28 benchmark tasks proposed in APE (Zhou et al., 2022); The results show that MoP surpasses comparable baselines and alternatives on the majority of the APE benchmark, obtaining exceptionally large gains (up to 43% absolute improvement) on the challenging ones. Our key contribution can be summarized as follows:

- We propose a Mixture-of-Prompt (MoP), a Mixture-of-Expert framework that divides the problem space into homogenous regions

- We extend the output space of the previous prompt optimization method to both instruction and demos.

- We show that the proposed two-step search algorithm, which leverages semantical similarity for demo assignment and routing function and region-based joint search for instruction assignment, achieves significant performance gains on the APE Benchmark.

## 2 RELATED WORK

**Prompt optimization for language generation.** Aligning a pretrained large language model (LLM) with human intentions is a crucial step toward unlocking the potential of large-scale text-based generative models (Ouyang et al., 2022; Schick et al., 2023; Kojima et al.). An effective line of training-free alignment methods is prompt optimization (PO) (Zhou et al., 2022). PO originated from in-context learning (ICL) (Dale, 2021), which is mainly concerned with various designs and arrangements of in-context demonstrations (Wei et al., 2022a; Yao et al., 2023). It later evolves into automatic prompt engineering, where powerful language models themselves are utilized to engineer prompts for various tasks (Zhou et al., 2022; Pryzant et al., 2023; Chen et al., 2023). The particular reason behind this emerging ability remains an open question. However, several preliminary studies have discovered a theoretical connection between ICL to traditional learning methods such as kernel Regression (Han et al., 2023), Bayesian Inference (Xie et al., 2021), and Gradient Descent (Dai et al., 2023; Von Oswald et al., 2023).

**Mixture of Experts Model.** The mixture of experts model (Jacobs et al., 1991; Jordan & Jacobs, 1994) is a classic paradigm of longstanding interest within the machine learning community. MoE structure was originally studied based on traditional machine learning models (Jordan et al., 1996; Collobert et al., 2001). Subsequently, it was extended to deep neural networks by (Eigen et al., 2013) to enhance its capacity in handling complex vision and speech problems. Following this development, there has been a proliferation of MoE layers integrated with various base neural network structures (Shazeer et al., 2017; Dauphin et al., 2017; Vaswani et al., 2017), leading to significant accomplishments in a wide range of language-related tasks. In recent years, efforts combining the MoE layer with various base network architectures have demonstrated remarkable successes in modeling natural languages. Our work extends this high-level paradigm developed in the architectural domain to the prompt optimization task, achieving substantial performance gain over other methods.

# 3 MoP: MIXTURE-OF-PROMPTS

## 3.1 FRAMEWORK OVERVIEW

We start by introducing key terminologies that will be used throughout the paper. We define a **Prompt** as the entire text preceding the question. We consider the setting where a prompt can be divided into two parts: (1). **Instruction**: a set of natural language sentences describing the task, and (2). **Demos**: a set of input-output pairs structured in a specific way to demonstrate how to solve a task. Below is an example prompt under this definition:

*Prompt = "Find the opposite words of the input. Input: Similar Output: Dissimilar ..."*

The mathematical formulation of a prompt ($P$) can be represented as follows (Xie et al., 2021):

$$P = [I, \tilde{\mathcal{D}}_{\text{train}}; x_{\text{test}}] = [I, x_1, \ y_1, \ o^{\text{delim}}, \ x_2, \ y_2, \ o^{\text{delim}}, \ ..., \ x_n, \ y_n, \ o^{\text{delim}}, \ x_{\text{test}}]. \tag{1}$$

Here, $I$ represents an instruction, $\tilde{\mathcal{D}}_{\text{train}} = \{(x_i, y_i)\}_{i=1}^n$ represents demos, which is the set of (input, output) sampled from the training demo dataset, $\mathcal{D}_{\text{train}}$, and $o^{\text{delim}}$ represents delimiter token.

**Automatic prompt optimization**  In recent studies, it has been empirically observed that task performance is significantly influenced by prompts (Wei et al., 2022b). This observation has given rise to the field of Prompt Engineering, aiming to find a prompt $P$ that, when provided to an LLM alongside a new input, denoted as $x_{\text{test}}$, enables the LLM to generate the corresponding output, represented as $y_{\text{test}}$. To automate this process, recent research formulates prompt optimization as the following search problem:

$$P^* = \arg \max_P \mathbb{E}_{(x_{\text{test}}, y_{\text{test}}) \sim \mathcal{D}_{\text{test}}} f(P, y_{\text{test}}), \tag{2}$$

which involves a task-specific scoring function denoted as $f(\cdot)$ (for more information, please refer to Appendix C). However, a limitation of existing auto-prompting methods is that they solely focus on searching for an optimal demo-free instruction. Note that this approach constrains the search space to a *single demo-free* instruction, i.e., $\tilde{\mathcal{D}}_{\text{train}} = \emptyset$ in Equation (1), thus we have $P = [I; x_{\text{test}}]$ in Equation (2). Such a limitation restricts their problem-solving capabilities, given that the problem space of NLP tasks can be too intricate to be adequately addressed by a *single demo-free* instruction.

**Mixture-of-Expert for prompt optimization.**  To address the aforementioned issue of existing auto-prompting methods, we expand the problem space coverage for automatic prompting by optimizing the Mixture of Prompts (MoP). To achieve this, we employ the Mixture of Experts (MoE) paradigm (Jacobs et al., 1991; Jordan & Jacobs, 1994) to partition the entire problem space into multiple homogeneous $C$ regions, each governed by a specialized expert, i.e., $P = \{P^{(c)}\}_{c=1}^C$. Within the MoE framework, prompt optimization (Equation (2)) transforms into an **expert assignment problem** that aims to search for the most suitable prompt for each expert, $P^{(c)*}$, with the ultimate goal of optimizing the performance of their mixture:

$$P^* = \arg \max_P \sum_{c=1}^C \mathbb{E}_{(x_{\text{test}}^{(c)}, y_{\text{test}}^{(c)}) \sim \mathcal{D}_{\text{test}}} f(P^{(c)}, y_{\text{test}}^{(c)}), \quad \text{where } P = \{P^{(c)}\}_{c=1}^C. \tag{3}$$

Here, $(x_{\text{test}}^{(c)}, y_{\text{test}}^{(c)})$ refers to the data point assigned to expert $c$ by the employed routing function during inference time (we explain it in more detail later in Section 3.2). Notably, our MoP framework expands the prompt for each expert to contain both the instruction and demos jointly optimized for each expert region in the problem space; $P^{(c)} = [I^{(c)}, \tilde{\mathcal{D}}_{\text{train}}^{(c)}; x_{\text{test}}]$ in Equation (3). Intuitively, concrete demos excel at defining fine-grained details and expertise (local information) matching the queries in a local region, whereas instructions provide general abilities and high-level explanations for solving tasks (global information). Inspired by this, we introduce a **two-phase search algorithm** that jointly optimizes (demos, instructions) pairs for each expert (detail in Section 3.2 and 3.3).

## 3.2 DEMO ASSIGNMENT

In our two-phase search algorithm, we initiate the process by grouping training demos, denoted as $\mathcal{D}_{\text{train}}$, into homogeneous clusters, with each cluster corresponding to a specific expert.

**Split demos to each expert based on their semantic similarity with K-Means clustering** Inspired by previous research demonstrating that the performance of in-context learning can be enhanced by retrieving demo samples that are semantically similar to the new input (Rubin et al., 2021; Han et al., 2023; Liu et al., 2021), we propose a clustering-based approach for demo assignment. Considering the approximation proposed in Han et al. (2023) where the in-context prediction of a test sample $x_i$ can be formulated as

$$\hat{y}_i = (\sum_j y_i K(x_i, x_j)) / (\sum_j K(x_i, x_j)), \tag{4}$$

where $\{(x_i, y_i)\}_{i=1}^n$ are demos used in the prompt and $K$ is some kernel that can be represented as $K(x_i, x_j) = \phi(x_i)^T \phi(x_j)$ with some embedding space $\phi(\cdot)$. Our goal is to divide the demos into $C$ groups $\{\mathcal{V}_1, \ldots, \mathcal{V}_C\}$ such that each group (expert) only uses its own demo. In this case, the same sample $x_i$'s prediction, assuming its in group $c$, will become

$$\bar{y}_i = (\sum_{j \in \mathcal{V}_c} y_i K(x_i, x_j)) / (\sum_{j \in \mathcal{V}_c} K(x_i, x_j)), \tag{5}$$

and the error $|\bar{y}_i - \hat{y}_i|$ is related to the sum of the kernel entries outside the cluster $\sum_{j \notin V_c} K(x_i, x_j)$. Therefore, a good demo assignment algorithm will minimize the sum of between-cluster kernel values while keeping the clusters balanced, leading to the following clustering objective:

$$\min_{\{\mathcal{V}_1, \ldots, \mathcal{V}_C\}} \sum_{c=1}^{C} \frac{\sum_{i \in \mathcal{V}_c} \sum_{j \notin \mathcal{V}_c} K(x_i, x_j)}{|\mathcal{V}_c|}. \tag{6}$$

Based on the derivation in Appendix B, this is equivalent to the following clustering objective:

$$\min_{\{\mathcal{V}_1, \ldots, \mathcal{V}_C\}} \sum_{c=1}^{C} \sum_{i \in \mathcal{V}_c} \|\phi(x_i) - m_c\|^2, \quad m_c = \frac{1}{|\mathcal{V}_c|} \sum_{j \in \mathcal{V}_c} \phi(x_j), \tag{7}$$

which is exactly the objective function of K-means clustering in the embedding space $\phi(\cdot)$. In practice, we assume $\phi(\cdot) := \mathcal{E}_\theta(\cdot)$ is a mapping formed by a neural network encoder, and conduct K-means in such embedding space to cluster demos. We will denote $\tilde{\mathcal{D}}_{\text{train}}^{(c)} = \{x_i\}_{i \in \mathcal{V}_c}$ as samples assigned to cluster $c$.

**Identify the optimal number of experts using K-Means-Auto** In the previously discussed K-means clustering algorithm, we adhere to a fixed number of experts, which can result in inaccurate demo assignments. Therefore, we suggest to automatically determine the optimal number of experts for a given training demo dataset $\mathcal{D}_{\text{train}}$ using pre-defined criteria. Specifically, quality clustering is characterized by data points within a cluster being closely grouped together while being far from data points in other clusters. When solely relying on the objective in Equation (7) in clustering, there is an inclination to opt for a larger number of experts, which can lead us away from our objective of achieving the optimal clustering for the demo dataset. Therefore, we employ scaled inertia, as defined in Equation (8), as a criterion for identifying the optimal number of experts.

$$C* = \underset{C=C_{\min}, \ldots, C_{\max}}{\arg\min} \left( \min_{\{\mathcal{V}_1, \ldots, \mathcal{V}_C\}} \sum_{c=1}^{C} \sum_{i \in \mathcal{V}_c} \|\phi(x_i) - m_c\|^2 + \alpha C \right). \tag{8}$$

**Measure the similarity between demos and routing function** To measure the similarity between different demos, we map the input part of each demo to the embedding space provided by a text encoder $\mathcal{E}_\theta$ and compute the $l_2$ distance. Note that the choice of embedding space does not have to be the same as the API model; as long as the embedding space reflects the high-level semantic similarity between different demos, it can be used to effectively partition the problem space. We also compare other options in the ablation study. In addition, during inference, experts are selected using a routing function that measures the distance between the embedding vector of the new query input and the centroids of clusters assigned to each expert. The selected expert is responsible for instructing the LLM in generating a response to a new input query, denoted as $x_{\text{text}}$.

### 3.3 INSTRUCTION ASSIGNMENT

Given the set of demos assigned to each expert, we now discuss various methods to assign instruction to each demo set. We will start with the most straightforward idea and derive our final algorithm step by step.

**Independent Search** The most naive method is to search for the best instructions independently of the demos. Concretely, we may use existing automatized methods to propose the best instruction and assign it to all the experts. The underlying assumption is that the optimal instruction for each expert is identical. However, our empirical evidence disproved this assumption (Section 4.3). The rationale is as follows: While each expert acquires a different specialty from the assigned demos, they also process distinct blind spots in terms of their general task-solving ability. Therefore, they might require different instructions to compensate for their special needs.

**Joint Search** When taking the demo splits into account, we can search for the optimal (instruction, demos) pair, by evaluating their collective performance on the validation set. While this method allows the instruction to be conditioned on the demo clusters, evaluating the performance on the entire validation set results in a gap between search and inference. Using the entire validation set measures how well an expert (instruction, demos) performs on the full data distribution;

However, during inference, each expert is only responsible for predicting the data within their region. Our empirical also supports this analysis; we find that the performance of an expert between the full and local distribution is not necessarily aligned (Figure 3).

**Region-Based Joint Search** To alleviate the issue in an exhaustive search, we first route each input in the validation set to its experts, then perform a joint search on the optimal (instruction, demos) pair. We termed this method "Region-Based Joint Search (RBJS)". Algorithm 1 in the appendix summarizes the entire search process.

$$I^{(c)*} = \arg\max_{I_j} \mathbb{E}_{(x_{\text{valid}}, y_{\text{valid}}) \sim \tilde{\mathcal{D}}_{\text{valid}, c}} f([I_j, \tilde{\mathcal{D}}_{\text{train}}^{(c)}, x_{\text{valid}}; y_{\text{valid}}]) \quad \text{for } j = 1, \ldots, M. \quad (9)$$

## 4 Experiments

In this section, we systematically validate our MoP framework, which jointly searches for the optimal (instruction, demos) pair to divide the problem space into homogeneous regions. We begin by describing the experimental setup for evaluating our framework in Section 4.1. In Section 4.2, we first validate whether the proposed clustering algorithm, which adopts the K-Means-Auto clustering method in the embedding space, can group the demo samples with semantically similar meanings. In Section 4.3, we verify that different experts, clustered by the K-Means algorithm, exhibit different strengths for the same test inputs, thereby demonstrating the significance of expert assignment for each test input. In Section 4.4, we show the fluctuations in the rankings among clusters when fixed instructions change for fixed clusters, implying each cluster has a different optimal instruction. Lastly, in Section 4.5, following the previous auto-prompting baseline (Zhou et al., 2022), we evaluate our MoP framework on 28 instruction induction tasks proposed by Honovich et al. (2022), i.e., APE benchmarks (Hereafter, we refer to the set of NLP tasks, which is considered in the APE paper, as the "APE benchmarks". Please refer to Appendix F.4 for more details on APE benchmarks).

### 4.1 Experimental Setup

**Backgrounds on APE** APE (Zhou et al., 2022), which is closely related to our framework, is an auto-prompting method that utilizes a LLM to automatically design prompts. Specifically, APE randomly samples 5 demos from the training dataset and adopts the templates corresponding to *Generating Instructions* from Table 11, along with the sampled demos. It then feeds this prompt into LLM to generate a set of 20 candidate instructions. For more details, please refer to Appendix F.

**Settings** We follow the settings in the original APE paper (Zhou et al., 2022) with the following exceptions. (1). Our evaluation is conducted on OpenAI's latest GPT-3.5-Turbo-Instruct model, a cost-efficient (100x cheaper) replacement for the text-davinci model used in APE. We reran APE on this model. (2). For all our methods, we report the mean and standard deviation across 4 runs to account for the randomness in the search phase. Since our Region-Based Joint Search (RBJS) uses APE-proposed instructions as the candidate set, we save the results from APE runs and reuse them in RBJS to eliminate the randomness in prompt proposals (please refer to Appendix F.3). Furthermore, we compare our method against two baselines: APE and APE with Demos, and defer more baselines that partially use our method to the ablation study. In addition to the individual accuracy improvements for each task, we also assess the overall comparative performance between methods by reporting their average win rates across all tasks.

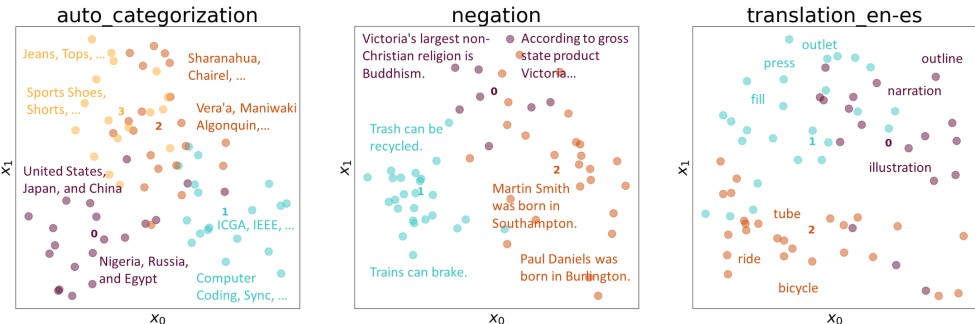

Figure 1: **Visualization results of demo clusters.** The results show the clustered demos after being mapped to the embedding space using a text encoder. Demos belonging to the same cluster are represented with the same color, and example texts within each cluster are shown accordingly.

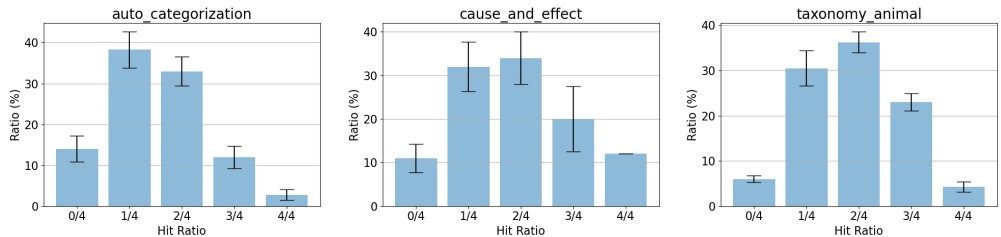

Figure 2: **Different experts have different strength.** Test inputs falling within the Hit Ratio range, except for 0/4 and 4/4, indicate that they are influenced by the specific expert they are assigned to, implying that each region's demos define distinct expertise to match queries within their respective local regions.

**Implementation Details**    We use the same set of hyperparameters throughout the study. For demo assignments, we use the default hyperparameter for K-Means-Auto without any modification. For instruction assignments, RBJS considers the top 4 prompts generated by APE.

### 4.2    VISUALIZATION OF DEMO CLUSTERS

In our MoP framework, the problem space is divided into homogeneous regions, with each region governed by a specialized expert. Building upon the theoretical connection between In-Context Learning and Kernel Regression, we begin by clustering a given set of demos into regions based on their semantic similarity. To achieve this, we first map the given demo sets into the embedding space using OpenAI's text-embedding-ada-002 model [1] as a text encoder ($\mathcal{E}$), which has shown remarkable performance on various NLP tasks, and subsequently apply the K-Means-Auto clustering algorithm with maximum number of clusters to be 4. To validate the ability of the proposed clustering algorithm to group semantically similar demos, we illustrate the visualization results of clustering in the embedding space followed by projection onto 2D with PCA in Figure 1. The results demonstrate that demo samples sharing semantically similar meanings are closely grouped together. For instance, in the auto_categorization task, which involves predicting characteristics shared by the given inputs, we observe that demos listing country names, demos related to computer science, demos representing extinct languages, and demos listing the types of apparel are respectively clustered together.

### 4.3    DIFFERENT EXPERTS HAVE DIFFERENT STRENGTH

In this section, we verify the impact of demo clusters on performance for each test input. To achieve this, we employ the K-Means method to form 4 clusters. In order to eliminate the impact of instructions on performance, all experts utilize only clustered demo samples as prompts, thereby restricting the output space to demos only. Subsequently, we calculate the Hit Ratio by counting the number of correctly answered experts out of the total number of experts ($C$) for each test input. If test inputs

---

[1]https://openai.com/blog/new-and-improved-embedding-model

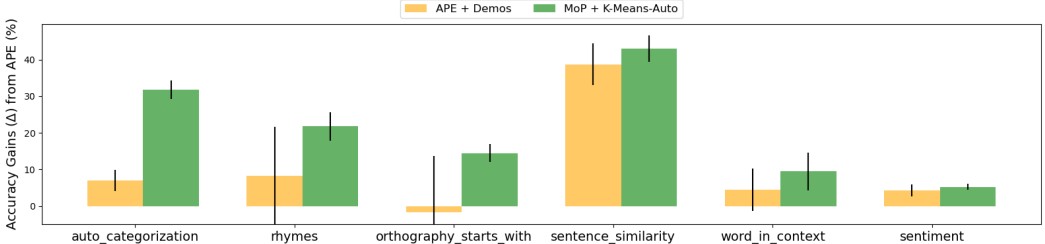

Figure 3: **Variations in Agent Rankings Across Different Instructions.** The results show the changing rankings among agents when different types of fixed instructions are provided to fixed agents. As depicted above, varying the fixed instructions results in fluctuations in the agent rankings, and there is no consistently high-performing agent.

Figure 4: **Accuracy Gains from APE on Instruction Induction tasks within the APE benchmark.** We provide accuracy gains ($\Delta$) from *APE + Demos* and *MoP + K-Means-Auto (Ours)* compared to *APE* for six tasks within the APE benchmark. Notably, our approach achieves an average accuracy gain of **20.96%** outperforming the average accuracy gain of 9.46% achieved by *APE + Demos* in these results. We run experiments with four different random seeds and report the mean and standard deviation.

yield Hit Ratios within the range other than $0/C$ and $C/C$, it indicates their sensitivity to the assigned expert. As depicted in Figure 2, we measure the Hit Ratios for the auto_categorization, cause_and_effect, and taxonomy_animal tasks and observe that, for each task, **83.25%**, **86.00%**, and **89.75%** of test inputs have Hit Ratio values that are neither 0 nor 1, respectively. This suggests that most test inputs are influenced by the type of clustered demos they are assigned, underscoring the fact that not all experts exert an equal impact on test inputs. Each region's demos define distinct expertise to match queries within their respective local regions.

## 4.4 NECESSITY OF REGION-BASED JOINT SEARCH

In this section, we investigate the necessity of jointly searching for the optimal (instruction, demos) pair. To accomplish this, we first create four clustered regions using the K-Means algorithm and maintain these clusters as fixed entities (i.e., Cluster ID: 1, ..., Cluster ID: 4). Next, for generating a set of instructions, we utilize the APE method (Zhou et al., 2022). For each task, we sample 5 input-output pairs from the training demos, generate 20 instruction proposals based on the sampled demos, and then select the top 4 instructions (Instruction 1 through Instruction 4) based on their evaluation performances. We subsequently use the demos specific to each region as prompts, in conjunction with the fixed instruction. Following this, we evaluate and rank these clusters based on their performance with test inputs. We repeat this process while varying the fixed instructions. As depicted in Figure 3, altering the fixed instruction results in changes in the rankings of the fixed clustered regions across all shown tasks. These results indicate that a single instruction is insufficient for all clustered regions, highlighting the need for distinct instructions for each region. This emphasizes the importance of a joint optimization scheme for the (instruction, demos) pair, which can lead to improved results.

## 4.5 RESULTS ON APE BENCHMARKS

In this section, we evaluate the effectiveness of our MoP framework on APE (Zhou et al., 2022) benchmarks by comparing it with other auto-prompting baselines.

*APE + Demos*, a variant of APE, incorporates demos sampled from the training demo dataset along with the best instruction selected by APE. Our *MoP + K-Means-Auto* employs K-Means-Auto to cluster training demo samples and determine the optimal instruction for each region. *MoP + K-*

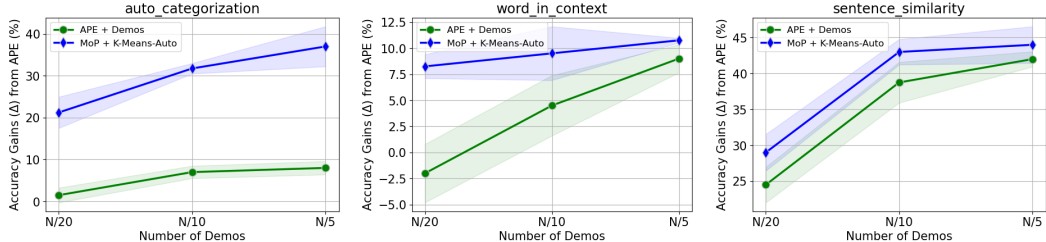

Figure 5: **Ablation study on different number of demos.** We conduct an ablation study on the number of demos. In this figure, $N$ on the x-axis represents the total number of training demos $\mathcal{D}_{\text{train}}$, and the y-axis represents the accuracy gain ($\Delta$) from APE for each method. *MoP + K-Means-Auto (Ours)* consistently surpasses the baseline methods *APE + Demos* across different numbers of demos in each task. We run experiments with four different random seeds and report the mean and standard deviation.

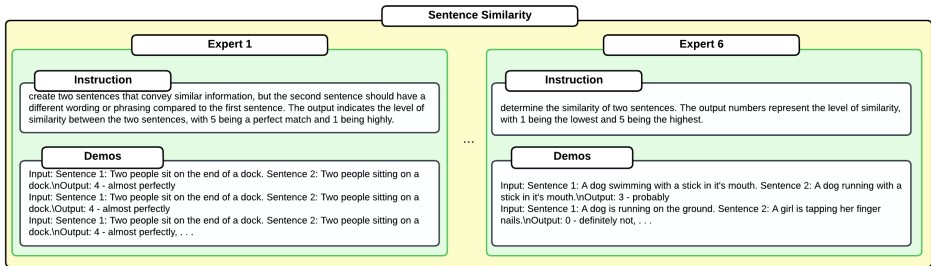

Figure 6: **The prompts for each expert found by MoP in the `sentence_similarity` task.** We provide examples of (instruction, demos) for each expert found by MoP for the `sentence_similarity` task. To see the complete results and results for other tasks, please refer to Appendix D.

*Means-Auto* uses the same number of demos as the *APE + Demos* method and selects the top 4 out of the 20 instructions generated by APE for the instruction candidates set.

We provide the results for 6 of the 28 tasks in Figure 4 (for results on the entire APE benchmark tasks, please refer to Table 2.). It's important to note that we present the performance gain achieved by both *APE + Demos* and *MoP + K-Means-Auto* methods compared to APE. As shown in Figure 4, *MoP + K-Means-Auto* demonstrates significant performance improvements over APE, achieving gains of **31.75%**, **21.75%**, **14.50%**, **43.00%**, **9.50%**, and **5.25%** across 6 tasks. In contrast, **APE + Demos** showed a performance gain of 7.0%, 8.25%, -1.75%, 38.75%, 4.5%, and 4.25%, indicating a smaller improvement compared to our method. These results demonstrate that our MoP framework, which divides the entire problem space into homogeneous regions, each governed by an expert with optimal (instruction, demos), effectively addresses complex NLP tasks. We provide prompts for each Expert found by MoP + K-Means-Auto for the `sentence_similarity` task in Figure 6.

## 5 ABLATION STUDY

In this section, we ablate the effect of different modules in the proposed MoP framework. We conduct the experiments on three challenging tasks from the APE benchmark: auto_categorization, word_in_context, and sentence_similarity the task throughout the section. All other settings are identical to the previous section.d

### 5.1 DIFFERENT NUMBER OF DEMOS

To establish a fair comparison between MoP and baselines, we set the number of demos to ($N_{\text{train}}/10$) in the main experiments, with $N_{\text{train}}$ representing the total number of training demos. In this section, we verify the performance of our method against the same baselines across different numbers of demos. As shown in Figure 5, our method consistently outperforms the baseline methods across various numbers of demos. Our method achieves a significantly higher accuracy than

*APE + Demos*, with an average improvement of **24.50%** (`auto_categorization`) / **5.66%** (`word_in_context`) / **3.58%** (`sentence_similarity`) across varying numbers of demos.

## 5.2 DIFFERENT CLUSTERING ALGORITHM

For demo assignment, we use K-Mean-Auto as the clustering algorithm, which automatically decides the best number of experts. The main intuition behind this choice is that more experts do not necessarily produce the best result. Here, we empirically validate this insight by comparing the performance of K-Means-Balanced and K-Means-Auto. Similar to Section 4.5, we set the number of experts K to $\frac{\text{Total \#demos in training set}}{\text{Max \#demos allowed in a prompt}}$, hence K-Means will always produce K clusters whereas K-Means-Auto will pick the best one from $2 \sim K$. As shown in Table 1, K-Means-Balanced performs similarly to Random Clustering, whereas K-Means-Auto significantly outperforms both.

## 5.3 DIFFERENT EMBEDDING MODEL

During demo assignment, we measure the semantic similarity of demos using $l_2$ distance in the embedding space. While our method is agnostic to the specific choice of embedding models, stronger text encoders perform better

Table 1: **Effect of different clustering algorithms (Top), embedding models (Middle), and prompt assignment algorithms (Bottom) on the performance of MoP in terms of prediction accuracy on APE Benchmark tasks.** The choice of models and algorithms in MoP is listed in the last row of every group. The best performance is achieved with K-Mean-Auto, text-embedding-ada, and region-based joint search.

| Clustering | similarity | context | auto_cate |
|---|---|---|---|
| Random | $36 \pm 3.4$ | $52 \pm 2.9$ | $33 \pm 1.5$ |
| K-Means-Balanced | $35 \pm 5.5$ | $58 \pm 3.4$ | $\mathbf{56 \pm 4.3}$ |
| K-Means-Auto | $\mathbf{43 \pm 3.5}$ | $\mathbf{59 \pm 5.2}$ | $54 \pm 2.5$ |

| Embed Model | similarity | context | auto_cate |
|---|---|---|---|
| GPT2 | $40 \pm 3.6$ | $56 \pm 3.6$ | $34 \pm 3.1$ |
| GPT2-Large | $39 \pm 2.6$ | $53 \pm 3.1$ | $50 \pm 2.7$ |
| Sentence-T5 | $\mathbf{43 \pm 3.0}$ | $58 \pm 5.9$ | $\mathbf{54 \pm 2.5}$ |
| Ada | $\mathbf{43 \pm 3.5}$ | $\mathbf{59 \pm 5.2}$ | $\mathbf{54 \pm 2.5}$ |

| Prompt Assignment | similarity | context | auto_cate |
|---|---|---|---|
| Independent | $41 \pm 4.2$ | $50 \pm 4.3$ | $45 \pm 5.7$ |
| Joint | $42 \pm 5.1$ | $\mathbf{59 \pm 1.6}$ | $46 \pm 2.1$ |
| Region-based Joint | $\mathbf{43 \pm 3.5}$ | $\mathbf{59 \pm 5.2}$ | $\mathbf{54 \pm 2.5}$ |

in identifying semantically similar demos. Here we examine how the strength of the embedding model affects the performance of MoP. We examine three text encoders of increasing sizes: GPT2, GPT2-Large, T5, and Ada-002. Unlike Ada and T5 which returns a single sentence-level embedding, GPT2 and GPT2-Large produce embeddings for each token. Therefore, for GPT2 variants, we obtain sentence-level embedding by averaging the embeddings of each token. Table 1's middle group summarizes the results.

## 5.4 DIFFERENT PROMPT ASSIGNMENT ALGORITHMS

We use the proposed Region-based Joint Search for prompt assignment. Recall that there are two main ideas behind Region-based Joint Search: 1). The optimal prompt for each expert might be distinct, therefore prompt assignment should be conditioned on the demo clusters (Joint Search); 2). The optimal prompt for each expert is evaluated only on the text points assigned to this expert (Region-based). As shown in the bottom group of Table 1, the proposed Region-based Joint Search outperforms both Joint Search and Independent Search, often by a large margin.

## 6 CONCLUSIONS

This work introduces the Mixture-of-Prompts (MoP) approach to enhance the performance of prompt optimization for Large Language Models. While existing methods search for a single instruction to prompt language models, MoP optimizes for a set of experts (prompts), each governing a specialized region of the problem space. This divide-and-conquer approach reduces the complexity associated with the task assigned to a single prompt, thereby substantially enlarging the problem space coverage. Within MoP framework, we further investigate various demo and instruction assignment methods for constructing the expert committee. Equipped with the proposed similarity-based demo assignment and region-based demo-instruction joint search, MoP substantially improves the performance over comparable methods over a diverse set of NLP tasks. We hope the proposed metho and associated findings could open up new possibilities for prompt optimization research.

**Limitations** We also include a discussion on the limitations of our method in Appendix G.

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
