---

**Algorithm 1:** Mixture of Prompts (**MoP**)

---

**Input:** Training demos $\mathcal{D}_{\text{train}} = \{(x_i, y_i)\}_{i=1}^{N_{\text{train}}}$, validation demos $\mathcal{D}_{\text{valid}} = \{(x_i, y_i)\}_{i=1}^{N_{\text{valid}}}$, text encoder $\mathcal{E}_\theta(\cdot)$, task-specific scoring function $f(\cdot) \rightarrow \mathbb{R}$.

/* Demo Assignment with K-Means-Auto.                                          */

**Input:** $\alpha$ in Equation (8), the minimum number of clusters $C_{\min}$, the maximum number of clusters $C_{\max}$

Compute $\boldsymbol{e}_i = \mathcal{E}_\theta(x_i)$ for $i = 1, \ldots, N_{\text{train}}$

Select the best $C$, which minimizes the scaled inertia score in Equation (8):

Clustering $\{\boldsymbol{e}_i\}$ into $C^*$ clusters using kmeans.

**Output:** Clustered demos $\{(x_i, y_i)\}_{i=1}^{|\tilde{\mathcal{D}}_{\text{train}}^{(c)}|}$    for $c = 1, \ldots, C^*$.

/* Instruction Assignment with Region-based Joint Search.                      */

**Input:** Set of Instruction Candidates $\{I_j\}_{j=1}^M$

Update the Region-based validation subset, $\tilde{\mathcal{D}}_{\text{valid}}^{(c)} \subset \mathcal{D}_{\text{valid}}$.

$\tilde{\mathcal{D}}_{\text{valid}}^{(c)} \leftarrow \emptyset$    for $c = 1, \ldots, C^*$

**for** $i = 1, \cdots N_{valid}$ **do**

$\quad \boldsymbol{e}_{\text{valid},i} = \mathcal{E}_\theta(x_{\text{valid},i})$

$\quad \hat{c} = \arg\min_{c=1,\ldots,C^*} d(\boldsymbol{e}_{\text{valid},i}, \boldsymbol{\mu}^{(c)})$

$\quad \tilde{\mathcal{D}}_{\text{valid}}^{(\hat{c})} \leftarrow \tilde{\mathcal{D}}_{\text{valid}}^{(\hat{c})} \cup \{(x_{\text{valid},i}, y_{\text{valid},i})\}$

Evaluate the score on the Region-based validation subset $\tilde{\mathcal{D}}_{\text{valid}}^{(c)}$ :

**for** $c = 1, \cdots, C^*$ **do**

$\quad I^{(c)*} = \arg\max_{I_j} \mathbb{E}_{(x_{\text{valid}}, y_{\text{valid}}) \sim \tilde{\mathcal{D}}_{\text{valid}}^{(c)}} f([I_j, \tilde{\mathcal{D}}_{\text{train}}^{(c)}, x_{\text{valid}}; y_{\text{valid}}])$    for $j = 1, \ldots, M$

**Output:** $\{I^{(c)*}\}_{c=1}^{C^*}$

---

# A    EXPERIMENTAL RESULTS FOR THE ENTIRE APE BENCHMARK TASK

Table 2: **Experimental results for the entire APE benchmark task.** We report the Execution Accuracy for each method on the entire APE benchmark task. We run 4 experiments and provide both the mean and standard deviation values. Due to innate randomness in ChatGPT API, we mark methods to be equivalent when their accuracy gap falls within 1%. The number of demos is set to $N_{\text{train}}/10$, where $N_{\text{train}}$ is the total number of training demos.

| Task | Execution Accuracy (%) | | | |
|------|------|------|------|------|
| | APE (Zhou et al., 2022) | APE + Demos | APE + K-centroids | MoP + K-Means-Auto |
| auto_categorization | 22.50±5.41 | 29.50±2.96 | 31.50±6.50 | **54.25±2.49** |
| rhymes | 30.50±14.77 | 38.75±13.44 | 37.00±15.17 | **52.25±3.96** |
| sentence_similarity | 0.00±0.00 | 38.75±5.67 | 25.00±6.75 | **43.00±3.54** |
| sentiment | 86.25±4.97 | 90.50±1.66 | 88.75±2.28 | **91.50±0.87** |
| word_in_context | 49.50±4.15 | 54.00±5.79 | 53.50±8.02 | **59.00±5.15** |
| larger_animal | **93.00±1.22** | **92.25±1.48** | 90.50±2.06 | **93.00±0.71** |
| informal_to_formal | 56.50±3.16 | **62.00±6.34** | 58.69±6.52 | **61.39±1.50** |
| orthography_starts_with | 54.75±10.23 | 53.00±15.48 | 65.25±8.87 | **69.25±2.49** |
| antonyms | 75.25±2.38 | 82.00±1.87 | 79.50±3.04 | **83.00±1.87** |
| second_word_letter | 59.00±0.00 | 80.25±19.83 | 80.25±18.75 | **81.25±18.27** |
| common_concept | 8.11±3.87 | **11.75±2.58** | 7.81±5.84 | **11.12±6.09** |
| cause_and_effect | 58.00±17.32 | 54.00±13.42 | **61.00±10.34** | 57.00±11.79 |
| translation_en-fr | 85.25±2.59 | **88.50±0.50** | 86.25±0.43 | **88.50±2.50** |
| diff | **100.00±0.00** | **100.00±0.00** | **100.00±0.00** | **100.00±0.00** |
| first_word_letter | **99.25±0.83** | **100.00±0.00** | **100.00±0.00** | **100.00±0.00** |
| letters_list | **100.00±0.00** | **100.00±0.00** | **100.00±0.00** | **100.00±0.00** |
| taxonomy_animal | **69.00±21.97** | 67.75±10.87 | **68.75±17.46** | **69.25±10.47** |
| negation | 78.00±3.00 | **87.50±1.50** | 82.00±1.87 | **87.25±1.48** |
| num_to_verbal | 98.00±0.00 | **100.00±0.00** | **100.00±0.00** | **100.00±0.00** |
| active_to_passive | 99.00±0.00 | **100.00±0.00** | **100.00±0.00** | **100.00±0.00** |
| singular_to_plural | 96.00±0.00 | **99.50±0.50** | 99.25±0.83 | **100.00±0.00** |
| sum | **100.00±0.00** | **100.00±0.00** | **100.00±0.00** | **100.00±0.00** |
| synonyms | **23.75±6.76** | 17.00±0.71 | 17.25±2.59 | 18.75±2.59 |
| translation_en-de | **82.50±1.66** | 82.00±1.22 | **83.25±2.17** | 82.25±1.09 |
| translation_en-es | 87.00±1.22 | 87.75±1.30 | **88.75±0.43** | **88.50±2.06** |
| auto_debugging | 28.12±5.41 | **43.75±6.25** | 40.62±5.41 | 40.62±10.36 |

We show the execution accuracy results for each method in the entire APE benchmark tasks in Table 2, excluding the two tasks for which the dataset has not been made publicly available: ascii and cs_algorithms.

## B    DERIVATION OF KMEANS OBJECTIVE

From equation 6 we have

$$\min_{\{\mathcal{V}_1,...,\mathcal{V}_C\}} \sum_{c=1}^{C} \frac{\sum_{i\in\mathcal{V}_c} \sum_{j\notin\mathcal{V}_c} K(x_i,x_j)}{|\mathcal{V}_c|}$$

$$= \min_{\{\mathcal{V}_1,...,\mathcal{V}_C\}} \sum_{c=1}^{C} \sum_{i\in\mathcal{V}_c} \left( \frac{\sum_{j\notin\mathcal{V}_c} K(x_i,x_j)}{|\mathcal{V}_c|} \right)$$

$$= \min_{\{\mathcal{V}_1,...,\mathcal{V}_C\}} \sum_{c=1}^{C} \sum_{i\in\mathcal{V}_c} \left( \frac{\sum_{j} K(x_i,x_j) - \sum_{j\in\mathcal{V}_c} K(x_i,x_j)}{|\mathcal{V}_c|} \right)$$

$$= \min_{\{\mathcal{V}_1,...,\mathcal{V}_C\}} \sum_{c=1}^{C} \sum_{i\in\mathcal{V}_c} \left( K(x_i,x_i) - \frac{\sum_{j\in\mathcal{V}_c} K(x_i,x_j)}{|\mathcal{V}_c|} \right) + \text{const}$$

$$= \min_{\{\mathcal{V}_1,...,\mathcal{V}_C\}} \sum_{c=1}^{C} \sum_{i\in\mathcal{V}_c} \left( K(x_i,x_i) - 2\frac{\sum_{j\in\mathcal{V}_c} K(x_i,x_j)}{|\mathcal{V}_c|} + \frac{\sum_{j,k\in\mathcal{V}_c} K(x_j,x_k)}{|\mathcal{V}_c|^2} \right)$$

$$= \min_{\{\mathcal{V}_1,...,\mathcal{V}_C\}} \sum_{c=1}^{C} \sum_{i\in\mathcal{V}_c} \left( \phi(x_i) - \frac{\sum_{j\in\mathcal{V}_c} \phi(x_j)}{|\mathcal{V}_c|} \right)^2$$

## C    SCORE FUNCTIONS

In the APE benchmark tasks we conducted, we evaluate the quality of prompts using a metric called *execution accuracy* proposed by Honovich et al. (2022). This metric assesses whether the desired output ($y_{\text{test}}$) is produced when a given prompt ($P$), along with the new input ($x_{\text{test}}$) is input into the LLM ($\mathcal{M}$). In most of the benchmark tasks, it is defined as $f(P, y_{\text{test}}) = \mathbb{1}[\mathcal{M}(P) = y_{\text{test}}]$. In certain tasks, a modified version of this metric is employed. For instance, it measures the proportion of correct answers within the total answer set. Please refer to Section 4.2 of Honovich et al. (2022) for further details.

# D   THE PROMPTS FOR EACH EXPERT FOUND BY MOP

We provide the (instruction, demos) pairs for each expert searched by our MoP framework in Table 3, Table 4, Table 5, and Table 6.

Table 3: **The prompts for each expert found by MoP in the sentence_similarity task.**

| | |
|---|---|
| **Task**: sentence_similarity
**Task Summary**: Rate the semantic similarity of two input sentences on a scale of 0 - definitely not to 5 - perfectly. | |

| | **Prompts** |
|---|---|
| Expert 1/6 | **Instruction**: create two sentences that convey similar information, but the second sentence should have a different wording or phrasing compared to the first sentence. The output indicates the level of similarity between the two sentences, with 5 being a perfect match and 1 being highly.
**Demos**: Input: Sentence 1: Two people sit on the end of a dock. Sentence 2: Two people sitting on a dock.\nOutput: 4 - almost perfectly\n\nInput: Sentence 1: Two people sit on the end of a dock. Sentence 2: Two people sitting on a dock.\nOutput: 4 - almost perfectly\n\nInput: Sentence 1: Two people sit on the end of a dock. Sentence 2: Two people sitting on a dock.\nOutput: 4 - almost perfectly, . . . |
| Expert 2/6 | **Instruction**: create two sentences that convey similar information, but the second sentence should have a different wording or phrasing compared to the first sentence. The output indicates the level of similarity between the two sentences, with 5 being a perfect match and 1 being highly.
**Demos**: Input: Sentence 1: Israel's Peres urges return to peace talks Sentence 2: Gonsalves keeps up pressure on Dom Rep\nOutput: 0 - definitely not\n\nInput: Sentence 1: Israeli, Palestinian negotiators quietly meet Sentence 2: Israeli, Palestinian negotiators meet in Jordan\nOutput: 4 - almost perfectly, . . . |
| Expert 3/6 | **Instruction**: compare the two sentences and determine how similar they are, giving a rating of 1 to 5 (with 1 being not similar at all and 5 being very similar).
**Demos**: Input: Sentence 1: Four men admit London Stock Exchange bomb plot Sentence 2: Four men admit London bombs plot\nOutput: 4 - almost perfectly\n\nInput: Sentence 1: Five men in tuxedos are walking down steps. Sentence 2: Five men in tuxedos walk down a set of steps\nOutput: 5 - perfectly, . . . |
| Expert 4/6 | **Instruction**: determine the similarity of two sentences. The output numbers represent the level of similarity, with 1 being the lowest and 5 being the highest.
**Demos**: Input: Sentence 1: A man is playing the trumpet. Sentence 2: Someone is playing with a toad.\nOutput: 1 - probably not\n\nInput: Sentence 1: A man is slicing a potato. Sentence 2: A woman is cutting a jalepeno.\nOutput: 1 - probably not, . . . |
| Expert 5/6 | **Instruction**:
**Demos**: Input: Sentence 1: U.S. Drone Kills Five Militants in Pakistan Sentence 2: Six Police Officers Killed in Attacks in Dagestan\nOutput: 0 - definitely not\n\nInput: Sentence 1: P.G. police seeking driver in crash that killed child Sentence 2: Police seek gunmen in New Orleans Mother's Day parade shooting\nOutput: 0 - definitely not, . . . |
| Expert 6/6 | **Instruction**: determine the similarity of two sentences. The output numbers represent the level of similarity, with 1 being the lowest and 5 being the highest.
**Demos**: Input: Sentence 1: A dog swimming with a stick in it's mouth. Sentence 2: A dog running with a stick in it's mouth.\nOutput: 3 - probably\n\nInput: Sentence 1: A dog is running on the ground. Sentence 2: A girl is tapping her finger nails.\nOutput: 0 - definitely not, . . . |

Table 4: **The prompts for each expert found by MoP in the auto_categorization task of the APE benchmark (Honovich et al., 2022).**

---

**Task**: auto_categorization
**Task Summary**: Categorize items based on a common theme or characteristic.

| | **Prompts** |
|---|---|
| Expert 1/8 | **Instruction**: group or categorize the items into related or similar groups. The input-output pairs all demonstrate this by organizing the items into categories such as journals, apparel, people shot by police, historical wars, presidents of the U.S, animals, computer science books
**Demos**: Input: Benin, Zambia, and Mauritius \nOutput: African countries\n\nInput: Nigeria, Russia, and Egypt\nOutput: countries with large population\n\nInput: Philippines, Brazil, and India \nOutput: countries with large population\n\nInput: Togo, Eritrea, and Burundi \nOutput: African countries . . . |
| Expert 2/8 | **Instruction**:
**Demos**: Input: Python, Cobol, and C#\nOutput: programming languages\n\nInput: Kotlin, Java, and Matlab\nOutput: programming languages\n\nInput: A History, a Theory, a Flood, How Google Works, and The Code Book: The Science of Secrecy from Ancient Egypt to Quantum Cryptography\nOutput: Computer Science books . . . |
| Expert 3/8 | **Instruction**: group or categorize the items into related or similar groups. The input-output pairs all demonstrate this by organizing the items into categories such as journals, apparel, people shot by police, historical wars, presidents of the U.S, animals, computer science books
**Demos**: Input: Sports Shoes, Shorts, and Sweaters\nOutput: Apparel\n\nInput: Jeans, Tops, and Suits\nOutput: Apparel, . . . |
| Expert 4/8 | **Instruction**: group similar items together based on a common characteristic or category. The output for each input is the category or characteristic that the items share.
**Demos**: Input: Sharanahua, Chairel, and Tafi\nOutput: extinct languages\n\nInput: Ngardi, Klallam, and Old Prussian \nOutput: extinct languages\n\nInput: Building of meteorological station, Manor Kozhin, and Watch tower\nOutput: tourist_attractions\n\nInput: The building of the gymnasium, Historical Reserve - the ruins of the mill them. KN Grudinina, and Chamber Stroganoff\nOutput: tourist_attractions\n\nInput: Vera'a, Maniwaki Algonquin, and Swoeng\nOutput: extinct languages, . . . |
| Expert 5/8 | **Instruction**:
**Demos**: Input: Mikhail Vrubel, Rene Magritte, and Amedeo Modigliani \nOutput: artists\n\nInput: James Bauduy, Patrick Bryant, and Elman Jerald Roberts \nOutput: people shot by police\n\nInput: Marie Curie, Albert Einstein, Mo Yan, Michael Houghton have all won\nOutput: the Nobel Prize\n\nInput: Tameka LaShay Simpson, Leslie Sapp III, and Edward Manning \nOutput: people shot by police\n\nInput: Augusta Savage, Louise Bourgeois, Mary Cassatt, and Elaine Sturtevant\nOutput: female artist\n\nInput: Mikhail Vrubel, Rene Magritte, and Amedeo Modigliani \nOutput: artists\n\nInput: James Bauduy, Patrick Bryant, and Elman Jerald Roberts \nOutput: people shot by police\n\nInput: Marie Curie, Albert Einstein, Mo Yan, Michael Houghton have all won\nOutput: the Nobel Prize, . . . |
| Expert 6/8 | **Instruction**: list things that fall into a certain category or theme.
**Demos**: Input: Azuchi-Momoyama period, Edo period, and Reiwa period \nOutput: periods of Japanese history\n\nInput: EverQuest II, Ikaruga, and State of Decay \nOutput: games on Steam\n\nInput: Wildlife Park 2 - Crazy Zoo, Glorkian Warrior The Trials Of Glork, and Software Inc. \nOutput: games on Steam\n\nInput: EverQuest II, Ikaruga, and State of Decay \nOutput: games on Steam\n\nInput: Wildlife Park 2 - Crazy Zoo, Glorkian Warrior The Trials Of Glork, and Software Inc. \nOutput: games on Steam, . . . |
| Expert 7/8 | **Instruction**: group or categorize the items into related or similar groups. The input-output pairs all demonstrate this by organizing the items into categories such as journals, apparel, people shot by police, historical wars, presidents of the U.S, animals, computer science books
**Demos**: Input: Rubella, Tetanus, and Varicella Hospitalizations\nOutput: vaccine preventable diseases\n\nInput: Plague, human, Dengue Virus Infection, and Cryptosporidiosis \nOutput: infectious diseases\n\nInput: Cysticercosis or Taeniasis, Anthrax, and Listeriosis \nOutput: infectious diseases\n\nInput: Ribosomes, plasmid, cell wall, and plasma membrane\nOutput: prokaryotic cells, . . . |
| Expert 8/8 | **Instruction**:
**Demos**: Input: TeenVogue.com, NME, and Tampa Bay Review\nOutput: the media\n\nInput: Government of Ontario News, WMTV, and La Crosse Tribune \nOutput: the media\n\nInput: WhoWhatWhy / RealNewsProject (blog), Today.com, and Rockford Register Star\nOutput: the media\n\nInput: TeenVogue.com, NME, and Tampa Bay Review\nOutput: the media\n\nInput: Physics Reports, Advanced Materials, and The Lancet Oncology \nOutput: top journals, . . . |

Table 5: **The prompts for each expert found by MoP in the negation task of the APE benchmark (Honovich et al., 2022).**

| | |
|---|---|
| **Task**: negation | |
| **Task Summary**: Negate the input sentence. | |

| | **Prompts** |
|---|---|
| Expert 1/3 | **Instruction**: negate the verb or verb phrase in the input sentence. 
 **Demos**: Input: Crawfordsburn is a village.\nOutput: Crawfordsburn is not a village.\n\nInput: Victoria's largest non-Christian religion is Buddhism.\nOutput: Victoria's largest non-Christian religion is not Buddhism. . . . |
| Expert 2/3 | **Instruction**: take a statement and negate it. 
 **Demos**: Input: Des Abbott was born in Darwin.\nOutput: Des Abbott was not born in Darwin.\n\nInput: Ellie Tesher was born in Toronto.\nOutput: Ellie Tesher was not born in Toronto.\n\nInput: Paul Daniels was born in Burlington.\nOutput: Paul Daniels was not born in Burlington. . . . |
| Expert 3/3 | **Instruction**: negate the verb or verb phrase in the input sentence. 
 **Demos**: Input: Trains can brake.\nOutput: Trains cannot brake.\n\nInput: Casper is a ghost.\nOutput: Casper is not a ghost.\n\nInput: Chlorine can be poisonous.\nOutput: Chlorine cannot be poisonous. . . . |

Table 6: **The prompts for each expert found by MoP in the orthography_starts_with task of the APE benchmark (Honovich et al., 2022).**

| | |
|---|---|
| **Task**: orthography_starts_with | |
| **Task Summary**: Extract the words starting with a given letter from the input sentence. | |

| | **Prompts** |
|---|---|
| Expert 1/3 | **Instruction**: identify the word that comes after the specified letter in each input sentence. 
 **Demos**: Input: Michael accidentally broke the glass. [m]\nOutput: michael\n\nInput: It was the man that bought the articles from him. [w]\nOutput: was\n\nInput: The fans were deliberately provoked by a rival group. [p]\nOutput: provoked\n\nInput: What Henri wants is the book which is on the top shelf. [s]\nOutput: shelf\n\nInput: This theorem will take only five minutes to prove. [m]\nOutput: minutes, . . . |
| Expert 2/3 | **Instruction**: identify the word that comes directly after the designated letter in the input sentence. So for example, for input [a], the output is "about" because it comes directly after the letter "a" in the sentence "John talked to Bill about the 
 **Demos**: Input: These are the things for which to be thankful. [a]\nOutput: are\n\nInput: Water filled the cup. [c]\nOutput: cup\n\nInput: for discussion of the same phenomenon in Russian. [r]\nOutput: russian\n\nInput: I know several people who she kissed. [i]\nOutput: i\n\nInput: I ate fruit [a]\nOutput: ate\n\nInput: We like our friends and they like their friends, too. [a]\nOutput: and\n\nInput: I know several people who she kissed. [s]\nOutput: several she, . . . |
| Expert 3/3 | **Instruction**: identify the word that comes after the specified letter in each input sentence. 
 **Demos**: Input: Being honest is not an easy task. [i]\nOutput: is\n\nInput: Almost any lawyer could answer that question. [t]\nOutput: that\n\nInput: They denied the claim that they should report only to us. [c]\nOutput: claim\n\nInput: John talked to Bill about the exam. [a]\nOutput: about\n\nInput: We expect the dentist to examine us. [d]\nOutput: dentist\n\nInput: He made a statement which everyone thought was really interesting and important. [t]\nOutput: thought\n\nInput: I wonder what John bought. [w]\nOutput: wonder what\n\nInput: June covered the baby with a blanket. [j]\nOutput: june\n\nInput: We persuaded the dentist to examine us. [w]\nOutput: we\n\nInput: So eminent a scholar as Dr. Lucille Hein was here. [e], . . . |

# E  ADDITIONAL EXPERIMENTAL RESULTS

## E.1  SUPER-NATURAL INSTRUCTIONS BENCHMARK

To further enhance the application value of our method, we validated our MoP framework on the Super-Natural Instructions benchmark (Wang et al., 2022b). The Super-Natural Instructions benchmark covers a range of tasks, such as commonsense classification and information extraction. While there are a lot of tasks encompassed within the Super-Natural Instruct benchmark, we validated our MoP framework on tasks related to code and mathematics.

Table 7: **Experimental results for the Super-Natural Instructions tasks.** We report the ROUGE-L score for each method. We run 4 experiments and provide both the mean and standard deviation values. Due to innate randomness in ChatGPT API, we mark methods to be equivalent when their accuracy gap falls within 1%. The number of demos is set to $N_{train}/10$, where $N_{train}$ is the total number of training demos.

| | Task | APE (Zhou et al., 2022) | ROUGE-L APE + Demos | APE + K-centroids | MoP + K-Means-Auto |
|---|---|---|---|---|---|
| Code | code_x_glue_information_retreival | 10.43±4.26 | 19.00±3.24 | 19.00±5.43 | **23.75±1.64** |
| | conala_list_index_subtraction | 38.32±10.01 | 40.81±7.13 | 45.13±7.46 | **49.98±6.30** |
| | conala_list_index_addition | 22.58±13.45 | 36.28±11.32 | **41.40±6.86** | **41.00±10.08** |
| | conala_calculate_mean | 35.00±0.71 | 31.75±1.09 | 35.25±3.27 | **37.25±1.48** |
| Mathematics | semeval_2019_task10_open_vocabulary_mathematical_answer_generation | 17.12±6.44 | **33.75±3.96** | **33.75±1.92** | **33.63±2.81** |
| | mathqa_gain | 14.99±1.58 | **26.25±4.26** | 22.75±3.42 | 24.00±5.61 |
| | mathqa_other | 13.34±0.91 | **29.25±1.64** | 23.50±2.96 | **29.00±3.67** |
| | mathdataset_answer_generation | 23.87±2.11 | 29.29±1.56 | 33.00±2.55 | **36.00±7.18** |
| | mathqa_geometry | 22.26±1.52 | 27.50±4.56 | **36.75±2.17** | 25.75±2.28 |
| | mathdataset_classification | 36.04±6.97 | 50.35±6.43 | 51.50±6.50 | **75.85±6.95** |
| | mathqa_general | 16.56±2.53 | 25.00±2.12 | 24.25±2.38 | **26.75±2.95** |
| | mathqa_probability | 16.47±1.07 | 26.75±2.28 | 24.25±2.68 | **30.25±0.83** |
| | semeval_2019_task10_closed_vocabulary_mathematical_answer_generation | 14.43 ±4.70 | **32.31±2.27** | 31.56±4.53 | 29.03±4.19 |

Initially, we investigated the performance of APE and conducted experiments on 13 tasks where APE struggled, i.e., tasks where APE's ROUGE-L score was below 50%. Given the diversity of the benchmark, we use the ROUGE-L score for reporting performance results in Table 7.

As described in Table 7, in 10 out of 13 tasks, our proposed MoP framework outperforms or is comparable to other baseline methods such as 'APE,' 'APE+Demos,' and 'APE+K-centroids'. These experimental results demonstrate the scalability of our proposed MoP framework not only on the APE benchmark but also across other tasks.

Furthermore, we also provide the prompts for each expert found by MoP for tasks 'mathdataset answer generation,' which involves mathematics, and the 'code x glue information retrieval,' related to coding in the Super-Natural Instructions benchmark, along with a task summary in Table 8, Table 9, and Table 10.

Table 8: **The prompts for each expert found by MoP in the mathdataset_answer_generation task of the Super-Natural Instructions benchmark (Wang et al., 2022b).**

---

**Task**: mathdataset_answer_generation
**Task Summary**: Answering multiple choices mathematical problem described with an open vocabulary.

| | **Prompts** |
|---|---|
| Expert 1/8 | **Instruction**: create various input-output pairs, so the outputs may not necessarily make sense in the context of the input. The instruction was likely for a math or programming problem involving solving equations or calculating probabilities.
**Demos**: Input: Convert 0.3324348 nanoseconds to seconds.\nOutput: 0.0000000003324348\n\nInput: Convert 0.3324348 nanoseconds to seconds.\nOutput: 0.0000000003324348\n\nInput: Convert 0.3324348 nanoseconds to seconds.\nOutput: 0.0000000003324348\n\nInput: Convert 0.3324348 nanoseconds to seconds.\nOutput: 0.0000000003324348, . . . |
| Expert 2/8 | **Instruction**: perform a calculation or solve an equation, and the output varies depending on the specific inputs given.
**Demos**: Input: Let p(s) be the first derivative of s**3/3 + s**2 + 3. Let v be p(-2). What is the units digit of 11/(-1)*(v + -1)?\nOutput: 1\n\nInput: Let s(g) = g**2 - 12*g + 3. Let i be s(12). Solve 1 = 2*p - 4*q + i, 0 = 4*q - 12 for p.\nOutput: 5\n\nInput: Suppose a - f + 0*f + 5 = 0, 5*f = -4*a + 25. Solve a*r + 2*r = 0 for r.\nOutput: 0\n\nInput: Suppose q + 3*d + 13 = -0*q, 2*q + d = -6. Let i = q + 10. Calculate the highest common divisor of 9 and i.\nOutput: 9, . . . |
| Expert 3/8 | **Instruction**: create various input-output pairs, so the outputs may not necessarily make sense in the context of the input. The instruction was likely for a math or programming problem involving solving equations or calculating probabilities.
**Demos**: Input: Two letters picked without replacement from x: 3, b: 2. Give prob of picking 2 x.\nOutput: 3/10\n\nInput: Four letters picked without replacement from w: 2, k: 1, o: 1, a: 2, b: 2. What is prob of picking 1 k, 2 a, and 1 b?\nOutput: 1/35\n\nInput: Four letters picked without replacement from s: 1, u: 1, v: 1, o: 2, f: 2, j: 1. Give prob of picking 1 v, 1 u, 1 j, and 1 s.\nOutput: 1/70, . . . |
| Expert 4/8 | **Instruction**: solve the given equations and provide the value of the specified variable in each output. The input-output pairs provided the results for different sets of equations and variables. In the first pair, the instruction asked to solve for s using the given equations and the value
**Demos**: Input: How many micrometers are there in 15/4 of a millimeter?\nOutput: 3750\n\nInput: What is eighteen fifths of a litre in millilitres?\nOutput: 3600\n\nInput: How many millimeters are there in 11/10 of a centimeter?\nOutput: 11\n\nInput: What is twenty-one eighths of a centimeter in micrometers?\nOutput: 26250, . . . |
| Expert 5/8 | **Instruction**: perform a calculation or solve an equation, and the output varies depending on the specific inputs given.
**Demos**: Input: What is the remainder when 46 is divided by (5*-5)/(19 - 20)?\nOutput: 21\n\nInput: Calculate (492/(-287))/((-30)/(-21)).\nOutput: -6/5\n\nInput: What is the value of 32/16*(-6)/(-22)?\nOutput: 6/11\n\nInput: What is the tens digit of (2 - (-12)/(-8))/(2/332)?\nOutput: 8\n\nInput: Calculate ((-12)/(-50))/((-20)/50).\nOutput: -3/5, . . . |
| Expert 6/8 | **Instruction**: create various input-output pairs, so the outputs may not necessarily make sense in the context of the input. The instruction was likely for a math or programming problem involving solving equations or calculating probabilities.
**Demos**: Input: Suppose -2*c + 4*j - 30 = 2*j, -5*c = -2*j + 78. Let w be 3/12 + (-12)/c. Solve 2*r = -w - 3 for r.\nOutput: -2\n\nInput: Suppose 3*y = -v + 6, -v + 4 = v - 2*y. Suppose 0 = -3*a - 2*c + 1, -4*c + 1 = -3*a - 8*c. Suppose h = 3 + a. Calculate the remainder when h is divided by v.\nOutput: 1,, . . . |
| Expert 7/8 | **Instruction**: solve the given equations and provide the value of the specified variable in each output. The input-output pairs provided the results for different sets of equations and variables. In the first pair, the instruction asked to solve for s using the given equations and the value
**Demos**: What is 7/4 of a year in months?\nOutput: 21\n\nInput: What is five quarters of a century in months?\nOutput: 1500\n\nInput: How many centuries are there in 78115.17 years?\nOutput: 781.1517, . . . |
| Expert 8/8 | **Instruction**: create various input-output pairs, so the outputs may not necessarily make sense in the context of the input. The instruction was likely for a math or programming problem involving solving equations or calculating probabilities.
**Demos**: Input: Two letters picked without replacement from gga. Give prob of sequence ga.\nOutput: 1/3\n\nInput: Two letters picked without replacement from naioiaaiaaaia. Give prob of picking 1 n and 1 i.\nOutput: 2/39\n\nInput: Two letters picked without replacement from hppfa. Give prob of sequence hf.\nOutput: 1/20, . . . |

Table 9: **The prompts for each expert found by MoP in the code_x_glue_information_retreival task of the Super-Natural Instructions benchmark (Wang et al., 2022b).**

| | |
|---|---|
| **Task**: code_x_glue_information_retreival
**Task Summary**: Given a code, calculate the number of "for loops" in the cpp program. | |

| | **Prompts** |
|---|---|
| Expert 1/8 | **Instruction**: create a function that counts the number of prime factors of a given number. The input-output pairs show different implementations of this function, with different variable names and control structures. The output values represent the number of prime factors for the given input number.
**Demos**: Input: void f(int i,int m);\nint sum;\nint main()\n{\n int n,i,m,k;\n scanf(\"%d\",&n);\n for(i=0;i<n;i++)\n { sum=1;\n scanf(\"%d\",&m);\n f(2,m);\n printf(\"%d\\n\",sum);\n \n }\n}\n void f(int i ,int m)\n {\n int k,s;\n s=(int)sqrt(m);\n for(k=i;k<=s;k++)\n {\n if(m%k==0)\n {\n sum++;\n f(k,m/k);\n }\n }\n }\n }\nOutput: 2
\n\nInput: int x[1000]={0};\nvoid f(int m,int n,int l)\n{\n\tint i;\n\tfor(i=m;i<=n;i++)\n\t{\n\t\tif(n%i!=0) continue;\n, . . . |
| Expert 2/8 | **Instruction**: write a function or a set of functions that would count the number of prime factors in a given number and output the result. Each input-output pair provided a different number for the function to calculate.
**Demos**: Input: int fenjie(int a,int b)\n{\n\tint i,sign=0,k=0;\n\tif(a<b) return 0;\n\telse if(a==b) return 1;\n\telse\n\t{\n\t\tfor(i=2;i<=a-1;i++)\n\t\t{\n\t\t\tif(a%i==0)\n\t\t\t\tsign=1;\n\t\t}\n\t\tif(sign!=1) return 1;\n\t\telse \n\t\t{\n\t\t\tfor(i=b;i<=a;i++)\n\t\t\t{\n\t\t\t\tif(a%i==0)\n\t\t\t\t\tk+=fenjie(a/i,i)\n\t\t\t}\n\t\t\tk+=1;\n\t\t\treturn k;\n\t\t}\n\t}\n}\n\n\tint main()\n{ \n\tint n,a[100],i,k;\n\tscanf(\"%d\",&n);\n\n\tfor(i=1;i<=n;i++)\n\t{\n\n\t\tscanf(\"%d\",&k);\n\t\tprintf(\"%d\\n\",fenjie(k,2));\n\t}\n\treturn 0;\n}\n\nOutput: 3
\n\n Input: int fenjie(int n,int min)\n{\nint x=0,i;\nif(n==1)\nreturn 1;\nelse\n{\n\tfor(i=min;i<=n;i++)\n\t{\n\t\tif(n%i==0)\n\t\tx=x+fenjie(n/i,i);}\n}\nreturn x;}\nmain()\n{int shu;\n\tint number,count=1;\n\tscanf(\"%d\",&number);\n, . . . |
| Expert 3/8 | **Instruction**: create a program that counts the number of prime factors of a given number. The input-output pairs provided by the friend seem to be testing different cases and inputs to make sure the program is functioning correctly. The output in each case seems to be the correct
**Demos**: Input: \nint sum(int a,int i)\n{\n\tint ret=1;\n\tfor(;i<=sqrt((float)a);i++)\n\t{\n\t\tif(a%i==0)\n\t\t{\n\t\t\tret+=sum(a/i,i);\n\t\t}\n\t}\n\treturn ret;\n}\nint main()\n{\n\tint t,a,j;\n\tcin>>t;\n\t\n\tfor(j=0;j<t;j++)\n\t{\n\t\tcin>>a;\n\t\tcout<<sum(a,2)<<endl;\n\t}\n\treturn 0;\n}\nOutput: 2
\n\nInput: int sum=0;\nint main()\n{\n\tvoid yinshu(int,int), . . . |
| Expert 4/8 | **Instruction**:
**Demos**: Input: int sum=0;\nvoid recur(int N,int i)\n{\n if(N==1) sum++;\n while(i<=N)\n {\n if(N%i==0) \n\t\trecur(N/i,i);\n i++;\n }\n return ;\n}\nint main()\n{\n int t=0;\n cin>>t;\n for(int k = 0; k <t; k++)\n {\n int n=0;\n cin>>n;\n int i=2,res=1;\n for(int i =2; i <=n/2;i++)\n {\n if(n%i==0)\n {\n sum=0;\n recur(n/i,i);\n res+=sum;\n }\n }\n cout<<res<<endl;\n }\n return 0;\n}\n\nOutput: 2\n\nInput: int sum=0;\nvoid divide(int n,int a)\n{\n\tfor(int i=a;i<=n;i++)\n\t{\n\t\tif((n%i==0))\n\t\t{\n\t\t\tif(n/i==1)\n\t\t\t\tsum++;\n\t\t\telse \n\t\t\t\tdivide(n/i,i);\n\t\t}\n\t}\n}\nint main()\n{\n\tint N=0,n=0,i=0;\n\tcin>>N;\n\tfor(i=0;i<N;i++) \n\t{\n\t\tcin>>n;\n\t\tdivide(n,2);\n\t\tcout<<sum<<endl;\n\t\tsum=0;\n\t}\n\treturn 0;\n}\nOutput: 2, . . . |
| Expert 5/8 | **Instruction**: write a function or a set of functions that would count the number of prime factors in a given number and output the result. Each input-output pair provided a different number for the function to calculate.
**Demos**: Input: int k=0;\nint y;\nint a(int x,int i);\nvoid b(int x,int i);\nint main()\n{\n\tint n,i,x,z;\n\tscanf(\"%d\",&n);\n\tfor(i=1;i<=n;i++)\n\t{\n\t\tscanf(\"%d\",&x);\n\t\ty=x;\n\t\t printf(\"%d\\n\",a(x,2));\n\t}\n\treturn 0;\n}\nint a(int x,int i)\n{\n\tk=0;\n\tb(x,i);\n\treturn k;\n}\nvoid b(int x,int i)\n{\n\tfor(i=i;i<=x;i++)\n\t{\n\t\tif(x%i==0) b(x/i,i);\n\t}\n\tif(x==1) k=k+1;\n}\nOutput: 2\n\nInput: int a;\nint fj(int x, int y);\nint main()\n{\n\tint n;\n\tcin >>n;\n\tint i;\n\tfor (i = 1; i <= n; i++)\n\t{\n\t cin >>a;\n\t cout <<fj(a, 2) <<endl;\n\t}\n\treturn 0;\n}\nint fj(int x, int y)\n{\n\tint xnumber = 1;\n\tint i;\n\tint b = sqrt(x);\n\tif (x == 1) return 0;\n\telse \n\t\tfor (i = y; i <= b; i++)\n\t\t{\n\t\t\tif (x % i == 0) xnumber = xnumber + fj(x / i, i);\n\t\t}\n\t\treturn xnumber;\n}\nOutput: 2, . . . |

Table 10: **(The extension of Table 9) The prompts for each expert found by MoP in the code_x_glue_information_retreival task of the Super-Natural Instructions benchmark (Wang et al., 2022b).**

| | |
|---|---|
| **Task**: code_x_glue_information_retreival | |
| **Task Summary**: Given a code, calculate the number of "for loops" in the cpp program. | |

| | **Prompts** |
|---|---|
| Expert 6/8 | **Instruction**: create a function that counts the number of ways an integer can be expressed as a product of two or more positive integers. The input-output pairs provided test the function with different inputs and expect the number of expressions as the output. Based on these pairs, 
 **Demos**: Input: void fun(int m, int i); //????\nint num ;\nint main()\n{\n\tint n, i, a;\n\tcin>>n;\n\tfor(i = 0; i <n; i++)\n\t{\n\t\tcin>>a;\n\t\tnum = 0;\n\t\tfun(a, 2); //????\n\t\tcout<<num + 1<<endl; //???????a=a???????num??1\n\t}\n\treturn 0;\n}\nvoid fun(int m, int i)\n{\n\tint k = sqrt(m);\n\tfor(i = i; i <= k; i++)\n\t{\n\t\tif(m % i == 0)\n\t\t{\n\t\t num++; \n\t\t\tfun(m / i, i); //?????????\n\t\t}\n\t}\n}\nOutput: 2\n\nInput: void count(int, int);\nint k;\n\nint main()\n{\n\tint n, i, j, m;\n\tint num;\n\tcin >>n;\n\tfor (i = 0; i <n; i++)\n\t{\n\t\tk = 0;\n\t\tcin >>num;\n\t\tm = sqrt(num) + 1;\n\t\tfor (j = 2; j <m; j++)\n\t\t\tif (num % j == 0)\n\t\t\t{\n\t\t\t\tk++;\n\t\t\t\tcount(num / j, j);\n\t\t}\n\t\tcout <<k + 1 <<endl;\n\t}\n\treturn 0;\n}\nvoid count(int num, int i)\n{\n\tint j;\n\tint n = sqrt(num);\n\tfor (j = i; j <= n; j++)\n\t{\n\t\tif (num % j == 0)\n\t\t{\n\t\t\tk++;\n\t\t\tcount((num / j), j);\n\t\t}\n\t}\n}\nOutput: 3, . . . |
| Expert 7/8 | **Instruction**: create a program that counts the number of prime factors of a given number. The input-output pairs provided by the friend seem to be testing different cases and inputs to make sure the program is functioning correctly. The output in each case seems to be the correct 
 **Demos**: Input: int f(int a,int min){\n\tif(a<min)\n\t\treturn 0;\n\tint sum=1;\n\tfor(int i=min;i<a;i++){\n\t\tif(a%i==0)\n\t\t\tsum+=f(a/i,i);\n\t}\n\treturn sum;\n}\nint main(){\n\tint n;\n\tint a[100];\n\tscanf(\"%d\",&n);\n\tfor(int i=0;i<n;i++){\n\t\tscanf(\"%d\",&a[i]);\n\t\ta[i]=f(a[i],2);\n\t}\n\tfor(int i=0;i<n;i++){\n\t\tprintf(\"%d\\n\",a[i]);\n\t}\n\treturn 0;\n}\n\nOutput: 3\n\nInput: \nint f(int a,int min)\n{ \nif(a <min)\n{ \nreturn 0; \n} \nint result = 1; \nfor(int i = min;i<a;i++)\n{ \nif(a % i == 0)\n{ \nresult += f(a/i,i); \n} \n} \nreturn result; \n} \n\nmain()\n{\n int n;\n scanf(\"%d\",&n);\n int i;\n int a;\n int b;\n for(i=0;i<n;i++)\n {\n scanf(\"%d\",&a);\n int min=2;\n b=f(a,2);\n printf(\"%d\",b);\n printf(\"\\n\"); \n }\n }\n\nOutput: 2, . . . |
| Expert 8/8 | **Instruction**: create a program that counts the number of prime factors of a given number. The input-output pairs provided by the friend seem to be testing different cases and inputs to make sure the program is functioning correctly. The output in each case seems to be the correct 
 **Demos**: Input: \nint main ()\n{\n\tint factor(int n,int x);\n\tint n;\n\tint i,m;\n\tint a[100];\n\n\tscanf(\"%d\",&m);\n\tfor(i=0;i<m;i++)\n\t{\n\t\tscanf(\"%d\",&n);\n\t\ta[i]=factor(n,2);\n\t}\n\tfor(i=0;i<m;i++)\n\t\tprintf(\"%d\\n\",a[i]);\n\treturn 0;\n}\n\nint factor(int n,int x)\n{\n\tint i,f=0;\n\n\tif(n==1)\n f=1;\n\telse\n\t\tfor(i=x;i<=n;i++)\n\t\t\tif(n%i==0)\n\t\t\t\tf+=factor(n/i,i);\n\treturn f;\n}\nOutput: 3\n\nInput: int main()\n{\n\tint nFactor(int n, int border);\n\tint n;\n\tcin >>n;\n\tfor (; n >= 1; n–)\n\t{\n\t\tint a;\n\t\tcin >>a;\n\t\tcout <<nFactor(a, a) <<endl;\n\t}\n\treturn 0;\n}\n\nint nFactor(int n, int border)\n{\n\tif (n == 1) return 1;\n\telse\n\t{\n\t\tint i, sum = 0;\n\t\tfor (i = border; i >= 2; i –)\n\t\t{\n\t\t\tif (n % i == 0)\n\t\t\t\tsum += nFactor(n / i, i);\n\t\t}\n\t\treturn sum;\n\t}\n}\n\nOutput: 2, . . . |

# F  BACKGROUNDS ON APE

In this section, to facilitate readers' understanding, we provide a detailed explanation of APE (Automatic Prompt Engineering (Zhou et al., 2022)), which is closely relevant to our work.

## F.1  THE BACKGROUND BEHIND AUTOMATIC PROMPT OPTIMIZATION

To begin with, we aim to explain the background behind auto-prompting methods, including APE (Zhou et al., 2022). While recent LLMs have demonstrated their remarkable ability to solve tasks described by user instructions (Ouyang et al., 2022; OpenAI, 2023; Touvron et al., 2023; Peters et al., 2018; Devlin et al., 2018; Brown et al., 2020; Wei et al., 2022b), carefully crafted prompts are crucial for maximizing LLMs' problem-solving ability. However, this often involves laborious trial and error. Recent attempts automate this by using LLMs to design prompts with their language generation ability, addressing tasks given demo datasets. APE is one of these auto-prompting methods, which has empirically demonstrated that LLM-generated prompts are more effective than human-crated prompts in solving target tasks.

## F.2 EXPLANATION OF THE APE ALGORITHM

In this section, we provide a more detailed explanation of the APE method. In APE (Zhou et al., 2022), firstly, it leverages a pre-trained black-box LLM to propose a set of candidate instructions. Specifically, APE initially selects random demos utilized for proposing instructions and adopts the templates corresponding to 'Generating Instructions' from Table 11, along with the sampled demos, into [FULL_DEMOS]. It then feeds this prompt into LLM to generate a set of candidate instructions. As described in Zhou et al. (2022), we used 5 randomly sampled demos from the training demos dataset ($\mathcal{D}_{\text{train}}$) for [FULL_DEMOS], resulting in the generation of 20 candidate instructions. We use these instructions generated by APE as candidate initial instructions for our proposed Mixture-of-Prompts (MoP) framework. After generating a set of candidate instructions in this manner, APE evaluates these generated candidate instructions using the validation set ($\mathcal{D}_{\text{valid}}$). Subsequently, it utilizes the top-1 instruction, which is a single demo-free instruction, during the test phase. For a fair comparison, all methods, including our MoP method and the APE method, use the same training, validation, and test datasets provided in the APE paper.

Table 11: **The templates used in our experiments.** We provide templates used in each scenario in our experiments. "Generating instructions" refers to generating initial instructions with APE, while "Evaluation" denotes the inference time (validation or test phase). For the case of "Listing Demos," it refers to the template used when listing multiple demo samples.

| Usage | Template |
|---|---|
| Generating Instructions | I gave friends an instruction. Based on the instruction, they produced the following input-output pairs:\n\n [FULL_DEMOS] \n \n The instruction was to <COMPLETE> |
| Evaluation | Instruction: [PROMPT] \n \n Input: [INPUT] \n Output: <COMPLETE> |
| Listing Demos | Input: [INPUT]\n Output: [OUTPUT] \n \n Input: [INPUT]\n Output: [OUTPUT] … |

## F.3 IMPLEMENTATION DETAILS ON INITIAL INSTRUCTIONS

As we aforementioned, we use the instruction sets generated by APE as the initial instruction set. However, due to the randomness of APE, we observed that different instruction sets are generated each time, even within the same scenario. We found this to hinder a fair comparison among methods. Therefore, as a solution, we first generate instruction sets using APE, save them, and reuse them to enable a fair comparison among the methods. The term "prompts generated by APE" used in the main paper refers to the saved optimized prompts that APE outputs.

## F.4 APE BENCHMARK TASK

We refer to the set of NLP tasks considered in the APE paper as the "APE benchmark" throughout the paper. As shown in Table 12, the APE benchmark encompasses tasks covering various aspects of language understanding, ranging from lexical semantics-related tasks to sentence style and cause selection. Detailed descriptions for each task are provided in Table 12.

## G EXPERIMENTAL RESULTS ON CHAIN-OF-THOUGHT REASONING TASKS.

To further validate the effectiveness of our method on reasoning-dense tasks, we conduct experiments on few-shot Chain-of-Thought benchmark tasks (Wei et al., 2022a). The Chain-of-Thought benchmark covers a range of tasks that demand human-level reasoning ability to solve.

Initially, we investigated the performance of APE and conducted experiments on 13 tasks where APE struggled, i.e., tasks where APE's ROUGE-L score was below 50%. Given the diversity of the benchmark, we use the ROUGE-L score for reporting performance results in Table 7.

As described in Table 13, in 10 out of 13 tasks, our proposed MoP framework outperforms or is comparable to other baseline methods such as 'APE,' 'APE+Demos,' and 'APE+K-centroids'. These experimental results demonstrate the scalability of our proposed MoP framework not only on the APE benchmark but also across other tasks.

Table 12: **Descriptions on APE benchmark tasks.** Referring to (Zhou et al., 2022), we provide task names, task summaries, and example demos within each task.

| Task | Task Summary | Demo |
|---|---|---|
| auto categorization | Categorize items based on a common theme or characteristic. | Python, Cobol, and C → programming languages |
| rhymes | Write a word that rhymes with the input word. | sing → ring |
| sentence similarity | Rate the semantic similarity of two input sentences on a scale of 0 - definitely not to 5 - perfectly. | Sentence 1: A man is smoking. Sentence 2: A man is skating. → 0 - definitely not |
| sentiment | Determine whether a movie review is positive or negative. | The film is small in scope, yet perfectly formed. → positive |
| word in context | Determine whether an input word has the same meaning in the two input sentences. | Sentence 1: Approach a task. Sentence 2: To approach the city. Word: approach → not the same |
| larger animal | Write the larger of the two given animals. | koala, snail → koala |
| informal to formal | Rephrase the sentence in formal language. | Please call once you get there → Please call upon your arrival. |
| orthography starts with | Extract the words starting with a given letter from the input sentence. | The man whose car I hit last week sued me. [m] → man, me |
| antonyms | Write a word that means the opposite of the input word. | won → lost |
| second word letter | Extract the second letter of the input word. | cat → a |
| common concept | Find a common characteristic for the given objects. | guitars, pendulums, neutrinos → involve oscillations |
| cause and effect | Find which of the two given cause and effect sentences is the cause. | Sentence 1: The soda went flat. Sentence 2: The bottle was left open. → The bottle was left open. |
| translation en-fr | Translate the word into French. | time → temps |
| diff | Subtract the second number from the first. | 32 22 → 10 |
| first word letter | Extract the first letter of the input word. | cat → c |
| letters list | Break the input word into letters, separated by spaces. | cat → c a t |
| taxonomy animal | Write all the animals that appear in the given list. | cat, helicopter, cook, whale, frog, lion → frog, cat, lion, whale |
| negation | Negate the input sentence. | Time is finite → Time is not finite. |
| num to verbal | Write the number in English words | 26 → twenty-six |
| active to passive | Write the input sentence in passive form. | The artist introduced the scientist. → The scientist was introduced by the artist. |
| singular to plural | Convert the input word to its plural form. | cat → cats |
| sum | Sum the two given numbers. | 22 10 → 32 |
| synonyms | Write a word with a similar meaning to the input word. | alleged → supposed |
| translation en-de | Translate the word into German. | time → Zeit |
| translation en-es | Translate the word into Spanish. | time → hora |
| auto debugging | Produce a specific result or output given the code. | import numpy as np \n x = numpy.zeros(10) \n → NameError: name 'numpy' is not defined. |

## H  LIMITATIONS

To promote future exploration, we discuss two limitations of the proposed method. First, the K-Mean-Auto algorithm used in the demo assignment does not guarantee the balance of the resulting clusters. When a cluster receives demos that exceed the limit, we randomly discard them to meet the constraint. This operation might be suboptimal as it does not factor in their relative importance. Future work might explore various data selection methods for trimming the cluster size. Second, while the prompt assignment algorithm is conditioned on the assigned demo, the candidate prompts are proposed independently (via APE). Proposals that take the assigned demos into account can potentially generate prompts that more effectively cater to each expert's specialty, thereby further improving the capability of the entire mixture.

Table 13: **Experimental results on Chain-of-Thought (CoT) Reasoning benchmark task.** We report the Execution Accuracy for each method on CoT reasoning tasks. Note that the original CoT paper only provides a limited number of human-crafted demos (8); we instead sample demos directly from the dataset for all methods. We run 4 experiments and provide both the mean and standard deviation values. Due to innate randomness in ChatGPT API, we mark methods to be equivalent when their accuracy gap falls within 1%. The number of demos is set to $N_{\text{train}}/10$, where $N_{\text{train}}$ is the total number of training demos.

| Task | Execution Accuracy (%) | | |
|---|---|---|---|
| | APE (Zhou et al., 2022) | APE + Demos | MoP + K-Means-Auto |
| GSM8K | 44.50$\pm$5.12 | 70.50$\pm$2.18 | **74.25$\pm$0.83** |
| MultiArith | 82.25$\pm$13.55 | **97.00$\pm$0.00** | **97.50$\pm$2.29** |