# OpenReview forum: "Mixture-of-Experts in Prompt Optimization"
_ICLR.cc/2024/Conference — Submitted to ICLR 2024_

### Official Review · Reviewer_cYiT · 2023-10-29

**Soundness:** 3 good
**Presentation:** 2 fair
**Contribution:** 3 good
**Rating:** 5
**Confidence:** 3

**Summary:**

This paper proposes Mixture-of-Prompts (MoP) algorithm to automatically create a series of prompts to tailor the need of different test cases. MoP leverages the idea of Mixture-of-Expert paradigm to divide the problem space into homogeneous regions governed by different experts by constructing a specialized prompt for each region via demo assignment and instruction assignment. The paper conducts extensive experiments to study the effects of different searching algorithms and finds that Region-based Joint Search gives the best results. Empirical results and in-depth analysis demonstrate the effectiveness of the proposed method.

**Strengths:**

1. Applying the idea of divide-and-conquer in prompt design is interesting. Unlike some methods that require manual prompt engineering, the MoP algorithm proposed in this paper can be a very general framework and automates the prompt engineering process in a systematic way.
2. The experiments and analysis are extensive and well-designed. The analysis of different experts' performance justifies the motivation of this work and can offer insights to the community as currently, most of the works still use a unified prompt rather than taking the property of each test case into account.

**Weaknesses:**

1. Though the authors view this method as an instantiation of Mixture-of-Expert paradigm in prompt engineering, I think the idea is actually similar to the classic ensemble learning. Considering that the expert construction has two phases, i.e., demo assignment and instruction assignment, have you tried fixing one of the phase and using different algorithms to decide the other? In this way, we can still get different prompts, although they may not be considered as different experts. I'm curious about the results if we just ensemble the outputs of these different prompts.
2. Some parts of the paper are not clearly written. See my detailed questions below.

**Questions:**

1. In test time, do you run the test case using all the prompts or have a router to decide which prompt to use? For Mixture-of-Expert paradigm, router or routing algorithm is a crucial part. However, the paper doesn't discuss this part clearly. Also, from the optimization goal in Equation (3), it seems like a single test case needs to be run multiple times. If this is the case, I'm wondering if your proposed MoP method will have efficiency issue.
2. Where do the initial demonstration set and instruction set come from? I know this paper is largely based on APE but it's better to briefly discuss these important details to make the paper self-contained. "Since our Region-based Joint Search uses APE proposed
prompts as the candidate set, we save the results from APE runs and reuse them in RBJS to eliminate the randomness in prompt proposals." I would suggest defining RBJS first to make readers easier. Also, from this sentence, it seems like the initial instruction set does influence the performance, so it's better to explicitly mention this part and explain a bit.
3. About the theory part of this paper, how do you go from Equation (5) to Equation (6)?
4. The experiments are mostly conducted on tasks that have close-ended output. I'm wondering if the MoP algorithm can be applied to more open-ended tasks, such as long-form QA or creative writing.

---

> ### Author Response · Authors · 2023-11-19
> **Response to R4 - cYiT - Part 1/4**
>
> Thank you for your positive feedback. We hope the responses below could address your questions and concerns. If you have any further questions, please feel free to let us know, and we are more than happy to discuss with you!
>
> **Concern 1 - How about ensemble of different experts?**
>
> *”Though the authors view this method as an instantiation of Mixture-of-Expert paradigm in prompt engineering, I think the idea is actually similar to the classic ensemble learning. Considering that the expert construction has two phases, i.e., demo assignment and instruction assignment, have you tried fixing one of the phase and using different algorithms to decide the other? In this way, we can still get different prompts, although they may not be considered as different experts. I'm curious about the results if we just ensemble the outputs of these different prompts.”*
>
> **[Response]**
>
> Thank you for your interesting suggestion. Although you already know the difference between MoE and ensemble, please allow us to reemphasize it again in case some less-familiarized public readers stepped upon this thread:
>
> 1. Mixture of Experts (MoE):
>     - Idea: MoE combines multiple models, called "experts," each specializing in different regions or types of data.
>     - Usage: The model has a routing mechanism that decides which expert to use for a particular input. Therefore, the inference cost is identical to a single model.
> 2. Ensemble Methods:
>     - Idea: Ensemble methods involve training multiple models independently and combining their predictions.
>     - Usage: The final prediction is often made by averaging (for regression) or voting (for classification) the predictions of individual models. Therefore, the inference cost is K times of a single model, where K is the number of ensembles
>
> Back to your suggestion: The following Table R1 shows how ensembling different experts compares with using the routing function (ours). The experiment is conducted on the same set of datasets as our ablation study in Section 5. As we can see, the ensemble performs poorly. This result can be expected since different experts are supposed to have different strengths. The failure can also be evidenced by Figure 2, which shows that only a minor portion of the test data can be correctly predicted by more than half of the experts. If this experiment is not what you had in mind, then it is likely we misunderstood your comment. If so, please let us know, and we will be more than happy to address it!
>
> **<Table R1>**
>
> |  | Ensemble of all experts | Route to one expert |
> | --- | --- | --- |
> | sentence_similarity | 0.1525 (0.1771) | **0.4300 (0.0354)** |
> | word_in_context | 0.5275 (0.0776) | **0.5900 (0.0515)** |
> | auto_categorization | 0.0075 (0.0130) | **0.5425 (0.0249)** |

---

> ### Author Response · Authors · 2023-11-19
> **Response to R4 - cYiT - Part 2/4**
>
> **Question 1 - routing function in MoP**
>
> *”In test time, do you run the test case using all the prompts or have a router to decide which prompt to use? For Mixture-of-Expert paradigm, router or routing algorithm is a crucial part. However, the paper doesn't discuss this part clearly. Also, from the optimization goal in Equation (3), it seems like a single test case needs to be run multiple times. If this is the case, I'm wondering if your proposed MoP method will have efficiency issue.”*
>
> **[Response]**
>
> Thank you for your comments. As mentioned in Section 3.2, paragraph “Measure the similarity between demos and routing function”, we use the routing function in MoP. Specifically, we route a test input to the closest cluster centroid and use the expert associated with that cluster to prompt the LLM. Since the same routing function is used in Equation (3), each test case will only need to be run by a single expert. Therefore, the efficiency is identical to a single forward pass.
>
> We would like to summarize the pipeline to put everything in place:
>
> - **Demo Assignment.**
>
> Given the theoretical connection between in-context learning and kernel regression [1, 2, 3], we group demos into clusters based on their semantic similarity and assign a cluster to each expert. To be more specific, we first measure the similarity between given different demos by mapping the input part of each demo to the embedding space using a given text encoder. As shown in Figure 1 of the main paper, this embedding space reflects the high-level semantic similarity between different demos, which can be effectively used to partition the problem space given the theoretical connection between in-context learning and kernel regression.
>
> - **Instruction Assignment: Region-Based Joint Search (Equation 3).**
>
> After grouping demos into clusters, now we start with a set of experts, each assigned a subgroup of demos and a set of candidate instructions (we use the ones generated by APE for fair comparison). Given a labeled validation set of (input, output) pairs, we first route each input in the validation set to its expert utilizing the routing function described below. Now, each expert has 1) a subset of demos, 2) a set of candidate instructions, and 3) a slice of validation set (regions); we can then rank the instructions (with demo added) on this validation slice to obtain the best one for this expert (please refer to Section 3.3 of the main paper for a more detailed explanation).
>
> - **The routing function during the test phase.**
>
> After instruction assignment, during inference, **experts are selected using the routing function** that measures the distance between the embedding vector of the query input and the centroids of clusters assigned to each expert (the same routing function applies when slicing the validation set). **The single selected expert** is responsible for instructing the LLM in answering that input, denoted as $x_{\text{test}}$.
>
> As discussed in the line following Eq (3), $({x_{\text{test}}}^{(c)}, {y_{\text{test}}}^{(c)})$ refers to the data point assigned to expert $c$. In other words, each data point will **only get assigned to one single expert $c$, rather than all experts.** Since a single test case needs to be **run one time, the efficiency remains identical to a single forward pass.** The summation term in Eq (3) represents the objective of our MoP framework, aiming to optimize the performance of the prompt mixture as a whole.

---

> > ### Comment · Reviewer_cYiT · 2023-11-21
> >
> > I thank the authors for the detailed response, especially clarifying the routing mechanism. Your efforts are greatly appreciated.
> >
> > After understanding this part better, I resonate with W1 mentioned by Reviewer rsCB. I also read your response to his/her review and doubt those assumptions hold true only on tasks where semantically similar samples can imply the final label of the output - the tasks used in the experiments have this property in common. I doubt whether the proposed method is still effective on more complicated tasks, like math problems or reasoning-intense QA problems.

---

> ### Author Response · Authors · 2023-11-19
> **Response to R4 - cYiT - Part 3/4**
>
> **Question 2 - Background on APE**
>
> *”Where do the initial demonstration set and instruction set come from? I know this paper is largely based on APE but it's better to briefly discuss these important details to make the paper self-contained. "Since our Region-based Joint Search uses APE proposed prompts as the candidate set, we save the results from APE runs and reuse them in RBJS to eliminate the randomness in prompt proposals." I would suggest defining RBJS first to make readers easier. Also, from this sentence, it seems like the initial instruction set does influence the performance, so it's better to explicitly mention this part and explain a bit.”*
>
> **[Response]**
>
> Thank you for your suggestion. As you mentioned, we believe it would be beneficial to add details about what you mentioned. We have addressed your question as follows:
>
> - **Additional self-contained explanation of the APE method.**
>
> To make our paper more self-contained regarding the APE method, we have added an explanation about APE in Appendix Section F of the revised version. Furthermore, before delving into our experimental section, we have made a paragraph about the background on APE and briefly covered it in Section 4.1 of the revision. To save your time, we briefly explain the APE algorithm as follows:
>
> 1. First of all, APE randomly samples demos utilized for generating instructions from the training set ($\mathcal{D}_{\text{train}}$).
> 2. Then, APE adopts the template for generating instructions along with the randomly sampled demos (please refer to Appendix Table 11 for more details on the templates). It feeds this prompt into LLM to generate a set of candidate instructions.
> 3. APE evaluates these generated candidate instructions using the validation set ($\mathcal{D}_{\text{valid}}$).
> 4. Based on the validation performance, APE utilizes the top-1 instruction, which is a single demo-free instruction, during the test phase with the test dataset ($\mathcal{D}_{\text{test}}$).
>
> To ensure fair comparison among methods, all methods, including APE and our proposed MoP methods, use the same training, validation, and test splits, which are provided in the APE paper [4]. In addition, our MoP method utilizes the same instructions set generated by APE in step 2 above as the initial candidate instructions set.
>
> - **Defining the term RBJS before using it.**
>
> Thank you for your feedback. In Section 3.3, we discuss Region-Based Joint Search. While we referred to this method as Region-Based Joint Search, we did not provide the abbreviation in that section indeed. As you suggested, we updated the revision by adding the abbreviation (RBJS) to facilitate readers' understanding. Furthermore, in the ‘Settings’ paragraph of Section 4.1, where you mentioned, we updated to clarify what the abbreviation referred to before using the RBJS term. We believe that thanks to your feedback, the readability of our paper has improved.
>
> - **Further explanation on the influence of the initial instruction set.**
>
> Following your comments, we have added an explanation about the influence of the initial instruction set generated by APE in Appendix Section F.3 of the revised version. In addition, to save your time, we will provide further clarification below:
>
> As we mentioned in Section 4.1 of the main paper, we use the instruction sets generated by APE as the initial instruction set. To be more specific, as described in the APE paper, APE samples 5 input-output pairs from the training demos dataset and generates 20 instructions for each task. However, due to the randomness of APE, we observed that different instruction sets are generated each time, even within the same scenario. We found this to hinder a fair comparison among methods. Therefore, as a solution, we first generate instruction sets using APE, save them, and reuse them to enable a fair comparison among the methods.
>
> While we described APE's instruction set generation in Section 4.5, we acknowledge the benefit of offering a more in-depth explanation, especially regarding the influence of the initial APE-generated instruction set, earlier in the Experimental section to enhance readability. In response to your feedback, we have explicitly included details about the initial instruction's influence in Appendix Section F.3 and added a pointer to this part in Section 4.1. We appreciate your comments since we believe that your suggestion has contributed to enhancing our paper’s flow further.

---

> ### Author Response · Authors · 2023-11-19
> **Response to R4 - cYiT - Part 4/4**
>
> **Question 3 - eq 5 → 6**
>
> *”About the theory part of this paper, how do you go from Equation (5) to Equation (6)?”*
>
> **[Response]**
>
> Our goal is to have $|\bar{y}_i - \hat{y}_i|$ as small as possible, which is the difference between Eq (4) and Eq (5). By comparing Eq (4) and Eq (5), we can see that when
>
> $\sum_{j\notin V_c} K(x_i, x_j) = 0$, and the difference will increase as we have larger $\sum_{j\notin V_c} K(x_i, x_j)$. Therefore, our goal is to minimize $\sum_{j\notin V_c} K(x_i, x_j)$ for all $i$, which means we want:
>
> $min_{V_1, …, V_C} \sum_{i=1}^n \sum_{j\notin V_c} K(x_i, x_j) = min_{V_1, …, V_C} \sum_{c=1}^C \sum_{i\in V_c} \sum_{j\notin V_c} K(x_i, x_j)$
>
> However, the above optimization problem has a trivial solution: we can assign all samples to the same cluster to achieve the minimal objective (0), but this trivial solution is not useful for partitioning the problem. Therefore, a common way in the clustering literature is to add the cluster size of each cluster as a penalty, leading to the objective of Eq (6). We want to emphasize that there are multiple ways to achieve a balanced cluster, and Eq (6) is just one of the methods, which will lead to the connection to the K-Means objective in Eq (7).
>
> ---
>
> **Question 4 - Open-ended tasks**
>
> *”The experiments are mostly conducted on tasks that have close-ended output. I'm wondering if the MoP algorithm can be applied to more open-ended tasks, such as long-form QA or creative writing.”*
>
> **[Response]**
>
> Thank you for raising this great point. This is actually a limitation that the prompt optimization community has yet to address. In principle, the current prompt can be extended to open-ended tasks, providing that there exists a reliable quantitative evaluation method to score the answers (since existing prompt optimization is solving a search problem). However, it is debatable whether such a quantitative evaluation method exists right now, as evaluating open-ended QA or creative content generation is generally hard [5]. We feel this is a limitation shared by LLM research in general, not limited to prompt optimization. However, we are pleased to see early efforts that leverage GPT-4 or specifically trained LLM-judgers to replace human evaluators on open-ended tasks. Furthermore, we look forward to exploring prompt optimization for less structured tasks in the future as well.
>
> ---
> **References**
>
> [1] Han, Chi, et al. "In-Context Learning of Large Language Models Explained as Kernel Regression." *arXiv preprint arXiv:2305.12766* (2023).
>
> [2] Rubin, Ohad, Jonathan Herzig, and Jonathan Berant. "Learning to retrieve prompts for in-context learning." *arXiv preprint arXiv:2112.08633* (2021).
>
> [3] Liu, Jiachang, et al. "What Makes Good In-Context Examples for GPT-$3 $?." *arXiv preprint arXiv:2101.06804* (2021).
>
> [4] Zhou, Yongchao, et al. "Large language models are human-level prompt engineers." *arXiv preprint arXiv:2211.01910* (2022).
>
> [5] Chan et al. ChatEval: Towards Better LLM-based Evaluators through Multi-Agent Debate. Arxiv 2023

---

> ### Author Response · Authors · 2023-11-23
> **Theoretical justification + results on reasoning tasks**
>
> We are grateful for your response and would like to initiate a 2nd round of discussion regarding your doubts about the demo assignment. We respectfully argue that the demo assignment-based clustering, though arguably not optimal, is both theoretically grounded and empirically validated:
>
> - Theoretical justification of the clustering assumption:
>
> We motivate the choice of “embedding distance” directly from the theoretical results on how LLM performs In-context learning (i.e. prompting LLM with demos). Concretely, prior work [1] has demonstrated that the way LLM performs ICL is theoretically equivalent to Kernel Regression in the kernel embedding space of those demos. Under this theoretical framework, the optimal demo assignment can be reduced to the same as the objective of clustering in embedding space, as derived in Section 3.2. The implication of such collection of clustering to ICL is that: **suppose there are tasks where clustering-based demo assignment schema fails, then it indicates ICL would fail as well, which means that the current LLM is not capable of handling these tasks in the first place. Such cases would be beyond the scope of our paper.**
>
> - Empirical evidence:
>
> Apart from the indirect empirical evidence provided in [1], **we further verify the strength of MoP on Chain-of-Thought reasoning tasks (Appendix G), especially the multi-step math problem-solving tasks as you suggested.** The results are summarized in the table below. As we can see, 1. 2. MoP still achieves consistently strong improvement against APE.
>
> | CoT Tasks | APE | APE + Demos | MoP + KMean-Auto (ours) |
> | --- | --- | --- | --- |
> | GSM8K | 44.50±5.12 | 70.50±2.18 | **74.25±0.83** |
> | MultiArith | 82.25±13.55 | 97.00±0.00 | **97.50±2.29** |
>
> We hope that the above strong and direct evidence from both theoretical and empirical perspectives could clear your doubts. If they still exist, please let us know your counterarguments against those evidence.
>
> Best,
> MoP Authors

---

### Official Review · Reviewer_rsCB · 2023-10-30

**Soundness:** 3 good
**Presentation:** 3 good
**Contribution:** 2 fair
**Rating:** 3
**Confidence:** 4

**Summary:**

This paper proposed a demo assignment task and an instruction assignment task with MoE framework. By this means, they hope to enhance the prompt searching capability. Their experiments demonstrate the effectiveness of their method.

**Strengths:**

S1: I like their presentation and language writing, which is clear.
S2: they proposed two-step search algorithm, which leverages semantical similar- ity for demo assignment and routing function and region-based joint search for instruction assignment, achieves significant performance gains on the APE Benchmark.

**Weaknesses:**

W1: The idea of grouping training demos into homogeneous clusters, with each cluster corresponding to a specific expert, is disputable. This method actually assumes that all the potential demos or test queries can be all decomposed by these groups. This causes several problems: (1) how to make sure the given training demos are representative enough to cover all the cases? (2) how to make sure your clustering method is effective in making each cluster homogeneous? (3) how to make sure your generated groups are qualified to support better performance? (4) If each group corresponds to one expert, how to set an optimal number of experts for different clustering methods? How to make sure these experts are representative enough, how to achieve the balance between expert number and efficiency?

Here is a very simple solution to all the problems raised by W1: ignore/remove the clustering component, and treat all the demos as normal instances that can be represented by a K-dimensional latent space where each dimension corresponds to one expert. I would like to see the feedback from the authors.


W2: can I assume that the proposed INSTRUCTION ASSIGNMENT is just the combination of Independent Search and Joint Search? I thank the authors for their insights but it is too straightforward, and there exist more important problems that should be considered. For example, since each expert corresponds to one sub-instruction, there is a pressing need to make sure these instructions are heterogeneous. Achieving this goal is not easy, for example, maybe we should maximize KL divergence among pairs of sub-instructions. I didn’t find such effort in this paper.

W3: Overall, I thank the authors for their hard work in solving an interesting problem. But I am not surprised about their novelties.

**Questions:**

Q1: In Equation (3), if we let $C=1$, can I treat this equation as this: we use one region with the entire training demos to search a single instruction? So the insight of your Mixture-of-Expert for prompt optimization is: you think searching one single instruction is hard (no matter whether we have training demos), and you reduce the difficulties by splitting this problem into several sub-problems (searching the combination of multiple sub-instructions with demos)

If my understanding is correct, I think the following sentence in your paper is over-claimed: "a limitation of existing auto-prompting methods is that they solely focus on searching for an optimal demo-free instruction" (below equation 2). It seems that the above sentence should be replaced with something like: "a limitation of existing auto-prompting methods is that searching an optimal instruction is usually very hard"

Q2: W1

Q3: W3

---

> ### Author Response · Authors · 2023-11-19
> **Response to R3 - rsCB - Part 1/3**
>
> **Concern 1 - K-dimensional**
>
> *”The idea of grouping training demos into homogeneous clusters, with each cluster corresponding to a specific expert, is disputable. This method actually assumes that all the potential demos or test queries can be all decomposed by these groups. This causes several problems: (1) how to make sure the given training demos are representative enough to cover all the cases? (2) how to make sure your clustering method is effective in making each cluster homogeneous? (3) how to make sure your generated groups are qualified to support better performance? (4) If each group corresponds to one expert, how to set an optimal number of experts for different clustering methods? How to make sure these experts are representative enough, how to achieve the balance between expert number and efficiency? [Here is a very simple solution to all the problems raised by W1: ignore/remove the clustering component, and treat all the demos as normal instances that can be represented by a K-dimensional latent space where each dimension corresponds to one expert. I would like to see the feedback from the authors.”*
>
> **[Response]**
>
> Thank you for your questions. We will first share our interpretation of your suggested solution and then explain why we opt for clustering in MoP. Please do correct us if we misunderstood your points.
>
> - **K-dimensional soft experts**
>
> In the solution you suggested, each demo will be mapped by some encoder to a K-dimensional embedding, where each dimension represents an expert. This way, the K-dimensional embedding can be viewed as weights of a demo to those experts. However, it is non-trivial to use those weights. The rationale is as follows: During the inference time, those demos will be ordered sequentially into one piece of text (as the example in Section 3.1) into text, and then used to prompt the LLM to make a prediction on a new test input. It would be non-trivial to put a weight on each demo.
>
> - **Why clustering**
>
> As mentioned in the paper, the choice of using clustering for demo assignment is directly inspired by the rich theoretical and empirical findings from in-context learning literature showing that demos that are semantically closer to a test input can do better at solving it; It suggests solving each test input using the set of demos that are semantical similarity to it. This directly aligns with the objective of clustering, and the connection is also explained in our theoretical analysis in Section 3.2.
>
> - **Back to the questions you raised**
>
> In our opinion, the questions you raised (how to choose the number of clusters, how to guarantee clusters are representative enough, …) are challenging for the whole area of clustering. Since the main focus of our work is to demonstrate the great potential of the MoE paradigm in prompt optimization, we view K-Means clustering as one of the demo assignment options, rather than something we try to solve.
>
> Further, while demo assignment based on clustering is arguably not optimal, it is directly inspired by the rich prior theoretical and empirical findings, easy to use, and withstand our extensive empirical evaluation; furthermore, it is non-trivial to come up with alternatives.
>
> *“1) how to make sure the given training demos are representative enough to cover all the cases?”*
> This is a fundamental challenge in (automatic) prompt engineering: if only a few demos are given, we never know whether those are sufficient to cover all the cases. However, this is more like the fundamental limit posted by the data: One could argue that the same logic applies to any discriminative model in machine learning as well. But if the data is insufficient, both our method and original APE (as well as any existing auto-prompting methods) can cover those cases.
>
> ”*2) how to make sure your clustering method is effective in making each cluster homogeneous?”*
>
> This is the main objective of clustering — existing clustering algorithms try to find the partition by maximizing within-cluster similarities and minimizing between-cluster similarities. For example, K-Means and spectral clustering are both designed for this objective. This is also the main reason why we resort to existing clustering algorithms to find the partition since they are designed to make sure each cluster is homogenous.
>
> => continue on the next part

---

> ### Author Response · Authors · 2023-11-19
> **Response to R3 - rsCB - Part 2/3**
>
> *”3) how to make sure your generated groups are qualified to support better performance?”*
>
> This is again related to the objective of clustering algorithms such as K-Means. As they find a partition to maximize within-cluster similarities and minimize between-cluster similarities, we are sure that the partition found by clustering is better than the initial objective without clustering. This implies that our method, even if it may not necessarily achieve global optimal, should be better than the naive baseline without demo clustering. This is also empirically supported by the significant performance gain of our method over APE + Demos (i.e., demos fall into a single cluster) in Figure 4 and Table 2, as you also pointed out in the strength section.
>
> *”4) If each group corresponds to one expert, how to set an optimal number of experts for different clustering methods?”*
>
> This is a great point and, in fact, exactly the reason we use K-Means-Auto instead of vanilla K-Means. Unlike vanilla K-Means, where you need to manually set the number of experts/clusters, K-Means-Auto automatically picks the best number of experts based on Eq (8). We ablate this choice in Table 1, which shows that KMean-Auto achieves overall stronger results than vanilla K-Means, and also performs more consistently across tasks.
>
> ---
>
> **Concern 2 - Instruction Assignment should consider diversity**
>
> *“can I assume that the proposed INSTRUCTION ASSIGNMENT is just the combination of Independent Search and Joint Search? I thank the authors for their insights but it is too straightforward, and there exist more important problems that should be considered. For example, since each expert corresponds to one sub-instruction, there is a pressing need to make sure these instructions are heterogeneous. Achieving this goal is not easy, for example, maybe we should maximize KL divergence among pairs of sub-instructions. I didn’t find such effort in this paper.”*
>
> **[Response]**
>
> Thank you for your question and suggestion. Before responding to your question, we would like to revisit the terminology in 3.1 para 1 just in case there are any misunderstandings: a prompt is defined as a language instruction followed by a set of demos. In this sense, both demos and the instruction can be considered prompts and also used independently to prompt LLM to solve a task. Using only instruction to prompt a LLM is considered in APE, and using only demos to prompt a LLM is connected to the concept of “In-Context Learning”.
>
> Now, we would like to first address the concern on expert diversity, then clarify the region-based joint search algorithm for instruction assignment.
>
> 1. In constructing Mixture-of-Experts, the ultimate goal is to improve the performance of the mixture as a whole, as this is what we really care about. And diversifying the experts is certainly one of the (indirect) methods to achieve this. In fact, the first phase of our search algorithm exactly promotes expert diversity by clustering demos based on their semantic similarity. Another more direct method is to optimize the performance of MoE as a whole by searching over all possible combinatorics of the demos. But this would be computationally prohibitive for the demo assignment part, so we ended up with the indirect max-diversity method, exactly like your suggestion.
> 2. However, for instructions, we can afford to use the more direct method since the set of candidate instructions is much less than possible demo combinations. The region-based joint search is exactly for optimizing the mixture as a whole. Given the demo groups generated in the previous phase, we first route each input in the validation set to its closest expert, then use those routed subsets to rank the instructions for each expert. This way, the level of diversity is controlled by the search algorithm. Despite the simplicity, this algorithm performs well on diverse sets of tasks and is much strong than the alternatives, as ablated in Table 1.
> 3. To further support the direct search, we compare region-based joint search with picking instructions for each expert to promote maximal pairwise diversity in embedding space. The results are summarized in the Table below. Here, Diversity means assigning the most diverse set of instructions to the experts, RBJS + Diversity refers to using the most diverse instructions as the candidate for RBJS search (as opposed to the top-4 global instructions), and RBJS is the default algorithm. As shown in Table 3, RBJS outperforms other variants with diversity considered.
>
> |  | Diversity | RBJS + Diversity | RBJS (ours) |
> | --- | --- | --- | --- |
> | sentence_similarity | 0.3450 (0.0409) | 0.4100 (0.0458) | **0.4300 (0.0354)** |
> | word_in_context | 0.5200 (0.0158) | 0.5350 (0.1074) | **0.5900 (0.0515)** |
> | auto_categorization | 0.4475 (0.0311) | 0.4650 (0.0269) | **0.5425 (0.0249)** |

---

> ### Author Response · Authors · 2023-11-19
> **Response to R3 - rsCB - Part 3/3**
>
> **Question 1 - Further clarification on the motivation**
>
> *“Q1: In Equation (3), if we let C = 1, can I treat this equation as this: we use one region with the entire training demos to search a single instruction? So the insight of your Mixture-of-Expert for prompt optimization is: you think searching one single instruction is hard (no matter whether we have training demos), and you reduce the difficulties by splitting this problem into several sub-problems (searching the combination of multiple sub-instructions with demos). If my understanding is correct, I think the following sentence in your paper is over-claimed: "a limitation of existing auto-prompting methods is that they solely focus on searching for an optimal demo-free instruction" (below equation 2). It seems that the above sentence should be replaced with something like: "a limitation of existing auto-prompting methods is that searching an optimal instruction is usually very hard"*
>
> **[Response]**
>
> Thank you for your question. The motivation of MoP is not “searching for a single instruction is hard”. Instead:
>
> 1. “prompting LLM to solve a task with a single instruction significantly limits the problem-solving potential”, because the complex problem space (or data distribution) of a NLP task often might not be covered by a single demo-free instruction. This is regardless of whether this instruction is searched or human-crafted.
> 2. Since the existing prompt optimization methods restrict their output space to a single demo-free instruction, the discovered prompt naturally inherits the above limitation.
> 3. Therefore, we propose to leverage the Mixture of Experts paradigm, allowing multiple specialized prompts with instructions + demos to a task.
> 4. Then, we develop a two-phase search algorithm for constructing the mixture of experts, this includes how to.
>
> This is the first time that the Mixture-of-Experts paradigm has been extended from the architecture domain to prompt optimization tasks. We demonstrated substantial improvement with several baselines over a wide range of NLP tasks (APE Benchmark + the newly added SuperNI benchmark in rebuttal (please refer to Appendix Section E.1)). Certainly, we do not claim that the components of the current search algorithm are the optimal choice by any means. Rather, they are our initial answers to the raised questions, and we backed them with extensive empirical and theoretical analysis. We hope that our work can demonstrate the potential of using the Mixture-of-Experts paradigm to the prompt optimization community and as a starting point for future research.

---

### Official Review · Reviewer_RqkC · 2023-10-31

**Soundness:** 3 good
**Presentation:** 2 fair
**Contribution:** 2 fair
**Rating:** 5
**Confidence:** 4

**Summary:**

The choice of prompt has been shown to be critical for the performance of Large Language Models. Therefore, many recent works have explored how to improve prompting strategies, one of them being prompt optimization. However, this suffers from two problems: first, the optimized prompt only contains an instruction, not demos (few-shot examples), and second, one prompt might not be the best for an entire dataset. This paper proposes a mixture-of-experts prompt optimization strategy to solve these two problems. First, the set of all demos is clustered in embedding space. These clusters, which tend to be semantically meaningful, are then each used to produce an optimized prompt. Then at inference, a sample is associated with a cluster (by measuring distance to centroids in embedding space) and is prompted using the cluster's instruction and demos. The advantage of this strategy is that it utilizes demos, and the instruction in the prompt can include local information that is relevant to its cluster but not necessarily to the rest of the dataset. That is, the proposed instruction per cluster is dependent on the demos belonging to that cluster, so the two components of the prompt complement each other. The paper measures this approach on NLP tasks and finds that it can outperform APE significantly (which samples the demos to create one instruction-only prompt for the entire dataset).

**Strengths:**

Originality:
- While ideas around using multiple prompts exist such as PromptBoosting [1], the MOE format of prompt optimization has not been studied to my knowledge. It would be interesting to extend this work into a soft MOE where outputs when using different prompts are aggregated in a weighted fashion (I guess going from discrete clustering slightly back to this kernel regression format), or to use multiple embeddings (something like [2] comes to mind) to produce different sets of candidate clusters.

[1] https://arxiv.org/abs/2212.09257
[2] https://arxiv.org/abs/2307.11031

Quality:
- Strong experiments with impressive gap over other prompt optimization approaches

Clarity:
- Notation and equations were easy to follow.

Significance:
- A clean algorithm for developing sets of specialized prompts. I think this approach can be useful in the future as long as tasks can be broken down into reasonably balanced sub-domains/topics.

**Weaknesses:**

Quality:
- Based on the description in 4.4, I have trouble understanding how creating 20 instruction proposals and picking the top 4 instructions is equivalent to the objective in equation 9. This might just be a clarity issue though; are you actually only showing the prompt optimizer samples from a given cluster each time (this is what I would expect)? 4.4 makes it sound like you are generating cluster-agnostic instructions and choosing the top 4 based on average performance on the dataset, not on the cluster in particular.
- I had some concerns about the instructions made available in the Appendix. First, some of them are incoherent (like Expert 2/3 in Table 6), and some of them ask very different things, such as sentence_similarity expert 1/6 asking to create sentences, and expert 3/6 asking to rate sentences. Also, I expected that the main advantage of utilizing the cluster is for the optimized prompt to pick up on some cluster-specific information. For example in Table 4, 3 of the 8 experts have specifications being "journals, apparel, ..., computer science books". But expert 1/8's demos appear to be geographic, expert 3/8 is apparel, expert 7/8 is health/medicine. This goes against the intention that the instruction and demos should complement each other. That being said, these prompts might be a byproduct of the APE method not working well.
- Another shortcoming mentioned in the limitations is the lack of incorporating dependencies; each cluster is individually optimized (at least based on the algorithm description). I am curious if a boosting style approach could help with this, or if you can generate instructions for the kth cluster by conditioning on demos and instructions that do _not_ belong to that cluster (e.g., telling the prompt optimizer that the instruction should be different from these other ones).

Clarity:
- In section 4, APE---both the tasks and the method---is not introduced the first time it is mentioned. I did not have any idea of what to expect of the evaluation tasks, since there was no pointer to a description of them besides an explanation of auto_categorization in 4.2 and an overview of them in 4.5, long after the APE tasks were introduced.
- It also was not clear what the APE method is, resulting in some confusion of "APE benchmarks" versus "prompts generated by APE".

**Questions:**

- Can you clarify the APE tasks, APE method, and how equation 9 is implemented?
- Why do the instructions not complement the demos for each expert?

---

> ### Author Response · Authors · 2023-11-19
> **Response to R2 - RqkC - Part 1/3**
>
> Thank you for your positive feedback. We hope the responses below could address your questions and concerns. If you have any further questions, please feel free to let us know, and we are more than happy to discuss with you!
>
> **Concern 1 - Clarify the connection between top 4 instructions and Eq (9) (region-based search).**
>
> *”Based on the description in 4.4, I have trouble understanding how creating 20 instruction proposals and picking the top 4 instructions is equivalent to the objective in equation 9. This might just be a clarity issue though; are you actually only showing the prompt optimizer samples from a given cluster each time (this is what I would expect)? 4.4 makes it sound like you are generating cluster-agnostic instructions and choosing the top 4 based on average performance on the dataset, not on the cluster in particular.”*
>
> **[Response]**
>
> Certainly, we can provide further clarification.
>
> Firstly, creating 20 instructions and picking the top 4 is not the objective of Eq (9). Instead, the top 4 instructions become the domain/search space of the argmax. In other words, these top 4 prompts are “**candidates**” for the Region-based joint search.
>
> Secondly, your expectation is correct, Eq (9) is the optimization taking place at each cluster. Each cluster starts with a candidate pool of instructions (M=4 in our setting) and searches for the best one by solving Eq (9).
>
> To further address your concern, we would like to clarify the message of Section 4.4:
>
> - **The message of Section 4.4: Necessity of Region-based Joint Search (RBJS).**
>
> Experiment in 4.4 is purely designed for analytical purpose, to validate the necessity of Region-based Joint Search. In 4.4, we first use APE to generate 4 cluster-agnostic instructions, then rank their performance when added to each expert. If there exists a global best instruction, then we would expect this instruction to rank top on all clusters; however, this is not the case, indicating that it is necessary to search for the instruction most suitable for each cluster, respectively, which is where our RBJS comes into play.
>
> ---
>
> **Concern 2 - Some APE-generated instructions are poor.**
>
> *”I had some concerns about the instructions made available in the Appendix. First, some of them are incoherent (like Expert 2/3 in Table 6), and some of them ask very different things, such as sentence_similarity expert 1/6 asking to create sentences, and expert 3/6 asking to rate sentences. Also, I expected that the main advantage of utilizing the cluster is for the optimized prompt to pick up on some cluster-specific information. For example in Table 4, 3 of the 8 experts have specifications being "journals, apparel, ..., computer science books". But expert 1/8's demos appear to be geographic, expert 3/8 is apparel, and expert 7/8 is health/medicine. This goes against the intention that the instruction and demos should complement each other. That being said, these prompts might be a byproduct of the APE method not working well."*
>
> **[Response]**
>
> Thank you for your detailed observation. Before addressing your concerns, we would like to clarify how the “candidate instructions” for each expert are generated, just in case there is ever any misunderstanding:
>
> 1. We use APE to generate a list of candidate instructions by sampling subsets from all available demos in the training set. We find that APE can already generate diverse instructions in this manner as the randomness from both “sampling” and ChatGPT provides sufficient variance.
> 2. Then, we deploy the region-based joint search to find the best instruction from this candidate pool for each expert.
>
> Back to your concern: Yes, APE sometimes does fail in the first place. As you observed, on a few tasks, APE fails to generate even reasonable instructions. Since our RBJS uses instruction proposals generated by APE, we occasionally have to tolerate its failed instructions. However:
>
> 1. Incorporating demos in the prompt is complementary to the optimized instruction and can correct the mistakes made by the instruction-only prompt to some extent.
> 2. Our method does not specifically rely on APE. Replacing APE with a potentially better instruction induction method can alleviate this issue.

---

> ### Author Response · Authors · 2023-11-19
> **Response to R2 - RqkC - Part 2/3**
>
> **Concern 3 & Question 2 - Instructions are generated independently of the demos.**
>
> *”Another shortcoming mentioned in the limitations is the lack of incorporating dependencies; each cluster is individually optimized (at least based on the algorithm description). I am curious if a boosting style approach could help with this, or if you can generate instructions for the kth cluster by conditioning on demos and instructions that do not belong to that cluster (e.g., telling the prompt optimizer that the instruction should be different from these other ones).”*
>
> *”*Why do the instructions not complement the demos for each expert?*”*
>
> **[Response]**
>
> Thank you for the great suggestion. It could indeed be beneficial to allow the instruction generation to condition the state of other clusters. Though, it might be non-trivial to get the technical details right to achieve the desired improvement. Additionally, boosting seems to be a great option to experiment with. We believe this could be a solid future direction to explore!
>
> ---
>
> **Concerns 4, 5 - Background on APE**
>
> *”In section 4, APE---both the tasks and the method---is not introduced the first time it is mentioned. I did not have any idea of what to expect of the evaluation tasks, since there was no pointer to a description of them besides an explanation of auto_categorization in 4.2 and an overview of them in 4.5, long after the APE tasks were introduced. It also was not clear what the APE method is, resulting in some confusion of "APE benchmarks" versus "prompts generated by APE".*
>
> **[Response]**
>
> Thank you for your suggestion! We agree that it should be discussed more before introducing it. We have added a brief discussion on APE, both the tasks and the methods, in **Section 4.1**, with more details in **Appendix F** due to the space limit.
>
> In addition, to clarify the APE-related terms for potential readers of this thread:
>
> 1. “APE benchmarks” refer to the NLP tasks considered in the APE (Automatic Prompt Engineering) paper [1].
> 2. “prompts generated by APE” refers to the set of prompts produced by APE algorithm.

---

> ### Author Response · Authors · 2023-11-19
> **Response to R2 - RqkC - Part 3/3**
>
> **Question 1 - APE task and eq.9**
>
> *”Can you clarify the APE tasks, APE method, and how equation 9 is implemented?”*
>
> **[Response]**
>
> Certainly, we will clarify them.
>
> - **Clarify the APE method and APE tasks.**
>
> The brief explanation of the APE method is as follows:
>
> 1. First of all, APE randomly samples demos utilized for generating instructions from the training set ($\mathcal{D}_{\text{train}}$).
> 2. Then, APE adopts the template for generating instructions along with the randomly sampled demos (please refer to Appendix Table 11 for more details on the templates). It feeds this prompt into LLM to generate a set of candidate instructions.
> 3. APE evaluates these generated candidate instructions using the validation set ($\mathcal{D}_{\text{valid}}$).
> 4. Based on the validation performance, APE utilizes the top-1 instruction, which is a single demo-free instruction, during the test phase with the test dataset ($\mathcal{D}_{\text{test}}$).
>
> To briefly explain the APE benchmarks, the benchmarks encompass tasks covering various aspects of language understanding, ranging from lexical semantics-related tasks to sentence style and cause selection. Detailed descriptions for each task are provided in **Appendix Table 12**.
>
> For more details, please refer to the response in **Concerns 4, 5**.
>
> - **How Eq (9) is implemented?**
>
> We start with a set of experts, each assigned a subgroup of demos and a set of candidate instructions (we use the ones generated by APE for fair comparison).
>
> Given a labeled validation set of (input, output) pairs, we first assign each sample in the validation set to the best expert for it.
>
> Now, each expert has 1) a subset of demos, 2) a set of candidate instructions, and 3) a slice of validation set (regions); we then rank the instructions based on the performance of (instruction, demos) pair on the validation slice and select the best pair for each expert.
>
> ---
>
> **Question 2 - Instructions are generated independently of the demos.**
>
> *”*Why do the instructions not complement the demos for each expert?*”*
>
> **[Response]**
>
> Please refer to the response to **Concern 3**.
>
> **A suggestion provided in “strength” - Extend this work into a soft MOE**
>
> *“It would be interesting to extend this work into a soft MOE where outputs when using different prompts are aggregated in a weighted fashion (I guess going from discrete clustering slightly back to this kernel regression format), or to use multiple embeddings (something like [2] comes to mind) to produce different sets of candidate clusters.”*
>
> Thank you for sharing the intriguing idea of soft MoE and providing the two relevant references! We agree that this is also an interesting and solid future direction to explore.
>
> ---
> **References**
>
> [1] Zhou, Yongchao, et al. "Large language models are human-level prompt engineers." *arXiv preprint arXiv:2211.01910* (2022).

---

> > ### Comment · Reviewer_RqkC · 2023-11-22
> >
> > Thank you for addressing my questions. For concern 2, your response makes sense, but I think it would be worth further quantifying the extent to which you are dependent on your choice of APE as your base layer optimizer. In future revisions of this draft, I'd suggest experiments like replacing APE with another prompt optimization method (such as https://arxiv.org/abs/2309.03409 or https://arxiv.org/abs/2311.05661) or removing the demos from the prompt.

---

### Official Review · Reviewer_24Yo · 2023-11-01

**Soundness:** 3 good
**Presentation:** 3 good
**Contribution:** 3 good
**Rating:** 6
**Confidence:** 4

**Summary:**

This paper focuses on prompt optimization for applying LLM on downstream task with (input, output) samples. The authors propose a method called Mixture-of-Prompt (MoP) as a novel approach to obtain optimized prompts. In details, the method first clusters the downstream samples (demos) using K-Means-Auto and then performs region-based joint search to determine the prompt for each cluster. During inference, a new downstream example will first be routed into one of the learned cluster and then use the cluster's optimized prompt. Experimental results on APE show the advantage of this method. Analysis on cluster numbers, region-based joint search, the
heterogeneity between clusters as well as other perspectives are conducted.

**Strengths:**

1. The motivation is interesting and reasonable. Typically for a downstream task, a fixed task instruction sentence may not be optimal to LLM for every sample of this task. In most cases, a downstream task dataset can also be subdivided into subtasks requiring various instructions. From these perspective, applying a mixed-of-prompt strategy is natural.
2. The method is clear and easy to follow. A two-phased approach is proposed and each phase is relatively easy to implement.
3. The authors have provided sufficient analysis both qualitatively and quantitatively. Case study on APE subtasks is also provided, providing more clearer information on the differences between the clustered experts.

**Weaknesses:**

1. I think more benchmarks in addition to APE should be considered, especially some datasets derived from real-scene user logs (like ShareGPT), which will demonstrate the application value of MoP better.
2. More results on embedding modules other than GPT, as well as including other baselines will be much better.

**Questions:**

1. I am curious on what prompt groups will the MoP obtain on math (especially some hard and complex math tasks) and code related tasks. If some evidence can be provided, this will be very good.
2. In many cases, it is not very easy for a task to split the task instruction and the input independently. How to make MoP work in such cases?

---

> ### Author Response · Authors · 2023-11-19
> **Response to R1 - 24Yo - Part 1/3**
>
> Thank you for your positive feedback. We hope the responses below could address your questions and concerns. If you have any further questions, please feel free to let us know, and we are more than happy to discuss with you!
>
> **Concern 1 - Consider benchmarks other than APE’s**
>
> *“I think more benchmarks in addition to APE should be considered, especially some datasets derived from real-scene user logs (like ShareGPT), which will demonstrate the application value of MoP better.”*
>
> **[Response]**
>
> Thank you for acknowledging the interesting and reasonable motivation behind our approach, as well as recognizing that we have provided sufficient qualitative/quantitative analysis.
>
> As you suggested, validating the effectiveness across more tasks would further enhance our approach. An important aspect in evaluating prompt optimization methods is that existing prompt optimization methods focus on finding the best prompt to instruct a LLM to perform **discriminative** tasks. Therefore, it remains challenging to define and expand the current framework of prompt optimization (adopted by all existing methods) to enable creative **generation**, e.g., ShareGPT.
>
> Instead, we incorporated an additional Super-Natural Instructions benchmark [1] to further enhance the application value of our method. The Super-Natural Instructions benchmark covers a range of tasks, such as commonsense classification and information extraction. While there are a lot of tasks within the Super-Natural Instruct benchmark, we focus on tasks related to coding and mathematics, as you suggested in Q1.
>
> - **Experimental Results**
>
> We follow the benchmark’s provided settings and use the ROUGE-L score as evaluation metrics. Also, similar to Table 2, we consider two methods’ performance to be equivalent (“tie”) if their difference is less than 1%; this helps to account for the inherite randomness of ChatGPT’s generation. Also, the number of demos is set to $N_{\text{train}}/10$, where $N_{\text{train}}$ is the total number of training demos. As shown in Table R1 below, our proposed MoP framework achieves the best result among all baselines - 'APE,' 'APE+Demos,' and 'APE+K-centroids' (for further details of APE+K-centroids, please refer to the response of C2) on **10 out of 13 tasks**. The new results further validate the generality and superiority of MoP across benchmarks and baselines.
>
> **<Table R1>**
>
> |  |  |  |  |  |  |  |  |  |
> | --- | --- | --- | --- | --- | --- | --- | --- | --- |
> | Task | code_x_glue_information_retreival | conala_list_index_subtraction | conala_list_index_addition | conala_calculate_mean | semeval_task10 |  |  |  |
> | **APE** | 0.1043 (0.0426) | 0.3832 (0.1001) | 0.2258 (0.1345) | 0.3500 (0.0071) | 0.1712 (0.0644) |  |  |  |
> | **APE + Demos** | 0.1900 (0.0324) | 0.4081 (0.0713) | 0.3628 (0.1132) | 0.3175 (0.0109) | **0.3375 (0.0396)** |  |  |  |
> | **APE + K-centroids** | 0.1900 (0.0543) | 0.4513 (0.0746) | **0.4140 (0.0686)** | 0.3525 (0.0327) | **0.3375 (0.0192)** |  |  |  |
> | **MoP + K-Means-Auto** | **0.2375 (0.0164)** | **0.4998 (0.0630)** | **0.4100 (0.1008)** | **0.3725 (0.0148)** | **0.3363 (0.0281)** |  |  |  |
> |  |  |  |  |  |  |  |  |  |
> | Task | mathqa_gain | mathqa_other | mathdataset_answer_generation | mathqa_geometry | mathdataset_classification |  |  |  |
> | **APE** | 0.1499 (0.0158) | 0.1334 (0.0091) | 0.2387 (0.0211) | 0.2226 (0.0152) | 0.3604 (0.0697) |  |  |  |
> | **APE + Demos** | **0.2625 (0.0426)** | **0.2925 (0.0164)** | 0.2929 (0.0156) | 0.2750 (0.0456) | 0.5035 (0.0643) |  |  |  |
> | **APE + K-centroids** | 0.2275 (0.0342) | 0.2350 (0.0296) | 0.3300 (0.0255) | **0.3675 (0.0217)** | 0.5150 (0.0650) |  |  |  |
> | **MoP + K-Means-Auto** | 0.2400 (0.0561) | **0.2900 (0.0367)** | **0.3600 (0.0718)** | 0.2575 (0.0228) | **0.7585 (0.0695)** |  |  |  |
> |  |  |  |  |  |  |  |  |  |
> | Task | mathqa_general | mathqa_probability | semeval_task10 |  |  |  |  |  |
> | **APE** | 0.1656 (0.0253) | 0.1647 (0.0107) | 0.1443 (0.0470) |  |  |  |  |  |
> | **APE + Demos** | 0.2500 (0.0212) | 0.2675 (0.0228) | **0.3231 (0.0227)** |  |  |  |  |  |
> | **APE + K-centroids** | 0.2425 (0.0238) | 0.2425 (0.0268) | 0.3156 (0.0453) |  |  |  |  |  |
> | **MoP + K-Means-Auto** | **0.2675 (0.0295)** | **0.3025 (0.0083)** | 0.2903 (0.0419) |  |  |  |  |  |
>
> We have included the experiments of Table R1 in **Table 7 of Appendix E.1** in the revision.

---

> ### Author Response · Authors · 2023-11-19
> **Response to R1 - 24Yo - Part 2/3**
>
> ---
>
> **Concern 2 - Ablate on Non-GPT embedding models & add more baselines**
>
> *“More results on embedding modules other than GPT, as well as including other baselines will be much better.”*
>
> **[Response]**
>
> - **Embedding models**
>
>     Thank you for your suggestion. The embedding models we experimented with (GPT-2, GPT-2-Large, Ada) are indeed all from the GPT series. We further ablated Google’s Sentence-T5 - a powerful encoder-decoder model. The results show that Sentence-T5 performs similarly to Ada, much stronger than other open-sourced GPT variants. This result further supports that MoP does not depend on a specific embedding model. We have included this result in our Ablation (**Table 1 in Section 5.3**).
>
>
> | Task | GPT-2 | GPT-2-Large | Sentence-T5 | Ada |
> | --- | --- | --- | --- | --- |
> | sentence_similarity | 0.3950 (0.0357) | 0.3900 (0.0255) | **0.4250 (0.0296)** | **0.4300 (0.0354)** |
> | word_in_context | 0.5625 (0.0363) | 0.5275 (0.0311) | 0.5750 (0.0589) | **0.5900 (0.0515)** |
> | auto_categorization | 0.3375 (0.0311) | 0.4950 (0.0269) | **0.5450 (0.0206)** | **0.5425 (0.0249)** |
> - **More baselines**
>
> Following your comments, we introduced another baseline termed “APE + K-centroids”. Instead of randomly sampling a subset of demos as APE + Demos, APE + Kcentroids pick the most representative demos to form a single cluster. This is achieved by selecting the demos associated with centroids of clusters derived from K-Means-Auto. We compare this new baseline with our method, “MoP + K-Means-Auto” in Table 2, where we found that MoP + K-Means-Auto achieves the best result (including “tie”, i.e., a difference within 1%) on all but 2 tasks. These results further demonstrate that our MoP framework can consistently bring improvement. We updated this new baseline on **Table 2 in the Appendix** to further strengthen our paper.
>
> **<Table R2>**
>
> |  |  |  |  |  |  |  |  |
> | --- | --- | --- | --- | --- | --- | --- | --- |
> | Task | antonyms | cause_and_effect | common_concept | diff | first_word_letter | informal_to_formal | larger_animal |
> | **APE+K centroids** | 0.7950 (0.0304) | **0.6100 (0.1034)** | 0.0781 (0.0584) | **1.0000 (0.0000)** | **1.0000 (0.0000)** | 0.5869 (0.0652) | 0.9050 (0.0206) |
> | **MoP+K-Means-Auto** | **0.8300 (0.0187)** | 0.5700 (0.1179) | **0.1112 (0.0609)** | **1.0000 (0.0000)** | **1.0000 (0.0000)** | **0.6139 (0.0150)** | **0.9300 (0.0071)** |
> |  |  |  |  |  |  |  |  |
> | Task | letters_list | taxonomy_animal | negation | num_to_verbal | active_to_passive | singular_to_plural | rhymes |
> | **APE+K centroids** | **1.0000 (0.0000)** | **0.6875 (0.1746)** | 0.8200 (0.0187) | **1.0000 (0.0000)** | **1.0000 (0.0000)** | **0.9925 (0.0083)** | 0.3700 (0.1517) |
> | **MoP+K-Means-Auto** | **1.0000 (0.0000)** | **0.6925 (0.1047)** | **0.8725 (0.0148)** | **1.0000 (0.0000)** | **1.0000 (0.0000)** | **1.0000 (0.0000)** | **0.5225 (0.0396)** |
> |  |  |  |  |  |  |  |  |
> | Task | second_word_letter | sentence_similarity | sentiment | orthography_starts_with | sum | synonyms | translation_en-de |
> | **APE+K centroids** | 0.8025 (0.1875) | 0.2500 (0.0675) | 0.8875 (0.0228) | 0.6525 (0.0887) | **1.0000 (0.0000)** | 0.1725 (0.0259) | **0.8325 (0.0217)** |
> | **MoP+K-Means-Auto** | **0.8125 (0.1827)** | **0.4300 (0.0354)** | **0.9150 (0.0087)** | **0.6925 (0.0249)** | **1.0000 (0.0000)** | **0.1875 (0.0259)** | 0.8225 (0.0109) |
> |  |  |  |  |  |  |  |  |
> | Task | translation_en-es | translation_en-fr | word_in_context | auto_categorization | auto_debugging |  |  |
> | **APE+K centroids** | **0.8875 (0.0043)** | 0.8625 (0.0043) | 0.5350 (0.0802) | 0.3150 (0.0650) | **0.4062 (0.0541)** |  |  |
> | **MoP+K-Means-Auto** | **0.8850 (0.0206)** | **0.8850 (0.0250)** | **0.5900 (0.0515)** | **0.5425 (0.0249)** | **0.4062 (0.1036)** |  |  |

---

> ### Author Response · Authors · 2023-11-19
> **Response to R1 - 24Yo - Part 3/3**
>
> ---
>
> **Question 1 - Experts discovered on math and programming tasks**
>
> *“I am curious on what prompt groups will the MoP obtain on math (especially some hard and complex math tasks) and code-related tasks. If some evidence can be provided, this will be very good.”*
>
> **[Response]**
>
> Thank you for bringing it up. In **Tables 8, 9, and 10 of** **Appendix Section E.1**, we added the experts discovered by MoP for tasks 'mathdataset answer generation,' (mathematics) and 'code x glue information retrieval,' (coding) from the Super-Natural Instructions benchmark, along with a task summary.
>
> ---
>
> **Question 2 - Tasks where the instruction and input are coupled**
>
> *“In many cases, it is not very easy for a task to split the task instruction and the input independently. How to make MoP work in such cases?”*
>
> **[Response]**
>
> Thank you for this insightful question. Yes, current prompt optimization methods all require a labeled set of (input, output) pairs to begin with, and the goal is to discover a prompt applicable to all the inputs. The case where the instruction and input (query question) are coupled, as you mentioned, has not been studied in existing prompt optimization literature yet. We think the primary question would be how to define the prompt optimization task for those cases: Existing prompt optimization methods aim to search for a universal prompt (or a set of prompts for the MoP case) that can be re-used and generalized to more than one input. However, when an instruction is specifically coupled with only one input, every input would require a different “instruction”.

---

### Author Response · Authors · 2023-11-19
**Common reply to all reviewers**

Firstly, we would like to express our gratitude to all of our reviewers for their efforts and attention to details in reviewing our paper. We are particularly encouraged by the positive feedbacks we received, such as:

1. Novel method with interesting and reasonable motivation. (R1, R2, R4)
2. Good presentation and easy to follow (R1, R2, R3)
3. Clean algorithm and relatively easy to implement (R1, R2)
4. Extensive and well-design experiments, with impressive performance gain (R1, R2, R3, R4)

Secondly, we are also blessed with many creative suggestions that our reviewers provided, and we incorporated them into our revision. To avoid messing with any references in the rebuttal period, we append most of the new content in the appendix (marked blue), but will make efforts to move them to the main text later.

Lastly, our reviewers expressed some concerns in the weakness section, and many of the concerns are, in fact, asking for clarification. We will address every point mentioned individually below, and we hope our response can help you in finalizing the scores of our paper. If you have any other questions, please feel free to reply back, and we will answer them asap!

**Highlighted changes in the revision:**

1. [TEXT] + Background on APE, including a discussion on its formulation, method, and the tasks considered.
2. [TEXT] + More discovered experts on math and coding tasks
3. [EXP]  + Incorporate SuperNI tasks into our evaluation
4. [EXP]  + Ablation on more embedding models

Sincerely,

MoP Authors

---

### Author Response · Authors · 2023-11-23
**Extra results on multi-step reasoning tasks (CoT)**

Dear reviewers,

As suggested by R4 - cYiT, we further provide results on the Chain-of-Thought reasoning benchmark, with a focus on multi-step math problem-solving tasks, in Appendix G. The results suggest that MoP's strong performance against APE is still consistent. We hope that the extra results can further clear your doubt about the applicability of MoP to reason-intense tasks.

Best,

MoP Authors

---

### Meta-Review · Area_Chair_ezA2 · 2023-12-20

**Metareview:**

This paper proposes a method (Mixture-of-Prompt) to optimize prompts with demonstration examples.

Strengths:
1. Simple approach and clear description.
2. Experimental results on APE show the advantage of this method.

Weakness:
1. Limited evaluation (only APE task).
2. It is unclear how this method generalize if there are completely new type of problems that do not fall into any of existing clusters.

Some weakness raised by the reviewers are address in response (validity on other embeddings).

**Justification For Why Not Higher Score:**

There are clear weakness (limited evaluation and generalization).

**Justification For Why Not Lower Score:**

N/A

---

### Decision · Program_Chairs · 2024-01-16

Reject